# DISRetrieval: Harnessing Discourse Structure for Long Document Retrieval

## Abstract

Long document understanding has become increasingly crucial in natural language processing, with retrieval-based methods emerging as a promising solution to address the context length limitations of large language models (LLMs). However, existing approaches either treat documents as flat sequences or employ arbitrary chunking strategies, failing to capture the inherent discourse structure that guides human comprehension. We present DISRetrieval, a novel hierarchical retrieval framework that leverages linguistic discourse structure to enhance long document understanding. Our approach introduces three key innovations: (1) a discourse-aware document organization framework that utilizes rhetorical structure theory (RST) to create sentence-level hierarchical representations, preserving both semantic relationships and natural document flow; (2) an LLM-enhanced node representation technique that combines discourse structure with adaptive summarization to enrich tree nodes with contextual information; and (3) a hierarchical evidence retrieval mechanism that effectively selects relevant content while maintaining discourse coherence. Through comprehensive experiments on QASPER and QuALITY datasets, DISRetrieval demonstrates substantial improvements over existing methods in both token-level retrieval metrics and downstream question answering tasks. Our ablation studies confirm that incorporating discourse structure significantly enhances retrieval effectiveness across different document lengths and query types, validating the importance of linguistically-informed document representation in long-text understanding.

## 1 Introduction

Comprehending natural language documents is a fundamental task in natural language understanding [32]. With the increasing granularity and complexity of textual data, long document understanding has gained significant attention [3, 18]. Large language models (LLMs) have shown remarkable effectiveness for this task [8]. However, they still struggle with the challenges of memory constraints and restricted context lengths [4, 28]. Retrieval-based methods are a promising solution to these challenges [22, 15]. These methods first split long documents into small pieces and then use a retrieval model to select the most relevant evidence, thereby easing the burden of long-context reasoning for LLMs [39, 42]. The document splitting strategy is a critical factor that determines both the granularity of evidence selection and the effectiveness of the retrieval model.

The straightforward document splitting method is to segment a full document into a sequence of flattened pieces [21, 23]. However, it fails to capture the inherent structural relationships and semantic dependencies that exist within the long document, leading to suboptimal retrieval performance [26, 7]. To this end, several researchers have attempted various strategies to decompose long documents hierarchically, as illustrated in Figure 1. The most representative work is RAPTOR [31], which

constructs a hierarchical tree for document organization through bottom-up semantic clustering. Additionally, an alternative bisection tree approach can be used, which only builds hierarchies between adjacent pieces to retain the original order of segmented pieces. Preliminary studies show that these tree-structure decomposition strategies are more effective for long document understanding.

Despite significant achievements in tree-based document organization, these methods generally ignore the real linguistically-informed discourse structures. Consider a document where the first sentence serves as a topic statement, followed by contrasting ideas (sentences 2-3), parallel supporting evidence (sentences 4-5), and a concluding statement (sentence 6). None of the previous retrieval-based methods have aligned with these types of knowledge structures. These discourse relationships naturally capture how humans organize and comprehend information in documents, making them particularly suitable for long document retrieval [13, 9, 33]. This discourse-centric perspective not only aligns with human cognitive processes but also provides a more principled approach to document segmentation and representation compared to arbitrary chunking or purely semantic clustering methods [17, 12, 34].

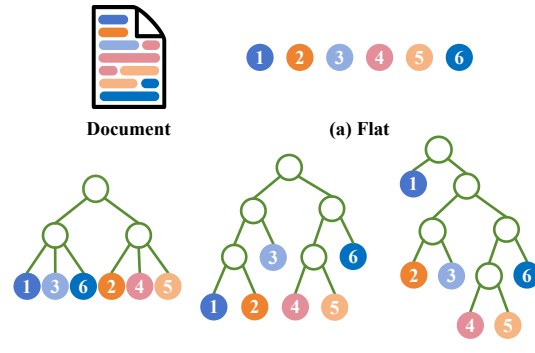

Figure 1: Comparison of document modeling approaches for long-text retrieval. Numbers (1-6) show sentence order in original document, with similar colors indicating semantic relationships. Four approaches are compared: (a) Flat sequential modeling, (b) Bottom-up semantic clustering of RAPTOR, (c) Bisection-based adjacent grouping, and (d) Our discourse-aware DISRetrieval that preserves both semantic and discourse structures.

In this work, we present the first approach that utilizes discourse information for retrieval-based long document understanding. Specifically, we propose DISRetrieval, which leverages rhetorical structure theory (RST) to organize long documents for retrieval. RST is a well-established discourse representation paradigm, using a hierarchical constituent tree to indicate the discourse structure of a document. The original RST exploits discourse-specific units as the bottom textual nodes. To better adapt our long document understanding, we ignore the inside-sentence discourse relationships, using only inter-sentence discourses. Thus, first we convert the original RST trees into sentence-level trees, and train an off-the-shelf RST model [38]. Second, we augment the non-leaf RST relation nodes with textual descriptions using an LLM agent. Finally, we construct an embedding tree based on the RST structure and employ it for query-based evidence retrieval to support LLM reasoning in long document understanding.

We conduct experiments on two benchmark datasets, QASPER [10] (a research paper QA dataset) and QuALITY [27] (a long-form reading comprehension dataset), to evaluate the effectiveness of our method. Results demonstrate that our approach significantly outperforms existing methods across various metrics. Specifically, our model achieves substantial improvements in both token-level retrieval metrics and downstream question answering tasks. Ablation studies validate the effectiveness of our discourse-aware tree construction and hierarchical retrieval mechanism, showing consistent gains over traditional chunking-based methods and flat retrieval approaches. These results demonstrate that our discourse-aware framework successfully captures both fine-grained semantic details and document-level patterns, while providing an effective solution for information propagation across different granularity levels in the hierarchy.

Our main contributions can be summarized as follows:

- We propose a novel discourse-aware hierarchical framework that leverages sentence-level discourse analysis to create semantically coherent document representations without relying on arbitrary chunking strategies.
- We develop an innovative bottom-up node representation technique that effectively combines discourse structure information with LLM-based semantic enhancement.
- Extensive experiments on two challenging datasets demonstrate significant improvements over existing methods, with up to 4% gains in retrieval metrics and consistent performance advantages across different model architectures.

## 2 Related Works

**Document Comprehending.**    Recent advances in natural language processing have revolutionized document processing and understanding [35, 40, 3]. While LLMs have demonstrated impressive capabilities, they face limitations due to context length constraints and memory bottlenecks [5, 8, 11, 1]. Traditional approaches often struggle with long-form content, as they typically operate on short text segments without considering the broader document context [23, 14, 30]. Dense retrieval methods, particularly DPR [21], have shown promising results on passage-level tasks but encounter challenges when scaling to long documents where understanding the overall structure becomes crucial [22, 15]. Several innovative approaches have emerged to address these limitations - SEER focuses on extracting key information without conventional chunking [43], while CFIC explores chunking-free in-context retrieval [29]. These developments highlight the ongoing evolution in document comprehension strategies, particularly for handling extended textual content [36, 20, 39].

**Retrieval-based Methods.**    The retrieval-based paradigm has emerged as a promising solution for long document understanding, fundamentally operating by first segmenting documents into manageable pieces before retrieving the most relevant evidence for downstream tasks [22, 21, 15, 19]. While basic approaches rely on flat sequential splitting, this method fails to capture the inherent structural relationships and semantic dependencies within documents [16, 3, 40]. More sophisticated tree-based organization methods have been developed, with RAPTOR leading the way through its implementation of bottom-up semantic clustering [31]. Alternative approaches include bisection-based methods that maintain local coherence by grouping adjacent segments [41, 18]. Recent work has also explored hybrid hierarchical retrieval (HHR) and dense hierarchical retrieval (DHR), establishing connections between document-level and passage-level representations for more contextually informed retrieval [2, 24]. However, these methods generally overlook the linguistic-informed discourse structures naturally present in documents [37, 25], leaving room for improvement in capturing document organization.

## 3 Methods

Retrieval-based methods have emerged as a promising solution for long document understanding, where the core idea is to first split documents into smaller pieces and then retrieve the most relevant evidence for downstream tasks. The most straightforward approach is to segment documents into flatten sequences. However, this approach cannot effectively capture the inherent structural relationships and semantic dependencies within long documents, resulting in suboptimal retrieval performance. Tree-based document organization methods provide enhanced document representation capabilities. Representative works like RAPTOR construct document hierarchies through bottom-up semantic clustering, while bisection-based approaches group adjacent segments to preserve local coherence. While these methods show improvements, they generally overlook the linguistic-informed discourse structures that naturally exist in documents.

We propose DISRetrieval, a discourse-aware retrieval framework that explores incorporating discourse structure into document retrieval. As a case study, we investigate the use of RST, a well-established discourse representation paradigm, to demonstrate how linguistic structures enhance document organization and retrieval effectiveness. By leveraging discourse relationships that reflect how humans organize and comprehend information, our framework maintains semantic coherence and structural integrity throughout the retrieval process, as illustrated in Figure 2. The algorithm of our approach is detailed in Algorithm 2 in Appendix C.

### 3.1 Discourse-Aware Tree Construction

The first stage of our framework aims to construct a hierarchical tree structure that captures semantic content and organizational structure of documents. We leverage discourse analysis to identify meaningful relationships between text segments and organize them into a coherent tree representation.

#### 3.1.1 RST Basics

RST provides a systematic framework for analyzing discourse structure in texts. An RST tree is a hierarchical representation where leaf nodes are text spans and internal nodes represent discourse

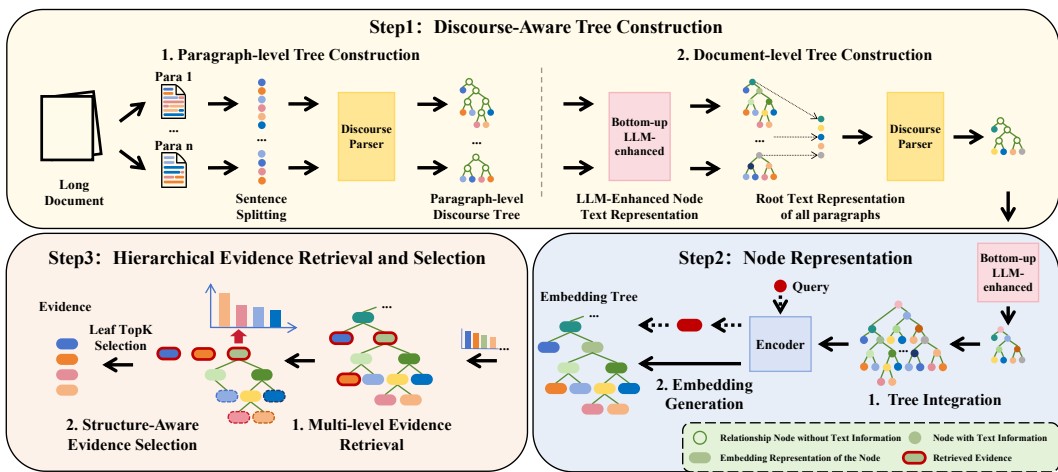

Figure 2: Overview of the DISRetrieval framework. The framework consists of three main steps: (1) Discourse-Aware Tree Construction that builds a hierarchical discourse structure through two phases: constructing paragraph-level discourse trees via discourse parsing, and generating document-level tree with LLM-enhanced node representations; (2) Node Representation that integrates the trees and converts text content into dense vector representations via an encoder; (3) Hierarchical Evidence Retrieval and Selection that performs multi-level evidence retrieval and structure-aware selection to identify relevant text segments.

relations between their children. These relations can be either mononuclear, where one span (the nucleus) is more central to the purpose of the writer than the other (the satellite), or multinuclear, where multiple spans are equally important. Common discourse relations include List, Contrast, Summary, and so on, which capture different ways that text segments relate to and support each other.

RST parsing is the task of automatically constructing RST trees from input text. Modern RST parsers typically employ neural networks to identify discourse boundaries and classify relations between text spans. These parsers first segment the text into elementary discourse units (EDUs), then incrementally build the tree structure by determining the relations between adjacent spans and combining them into larger discourse units. EDUs are the minimal building blocks in RST analysis, typically representing sub-sentence segments such as clauses. These fine-grained units serve as the leaf nodes of RST trees and form the basis for identifying discourse relationships in text.

### 3.1.2 Sentence-level Adaptation

While EDU-level discourse analysis provides fine-grained discourse structure, operating at the sentence level offers several advantages for long document understanding. First, sentences represent complete semantic units that maintain better coherence and interpretability compared to sub-sentence EDUs. Second, sentence-level segmentation significantly reduces computational complexity while preserving essential discourse relationships. Third, sentences align better with downstream retrieval tasks where the goal is to identify relevant text segments that can stand independently as evidence.

To enable sentence-level discourse analysis, we train a specialized discourse parser through the following process: (1) We merge EDUs belonging to the same sentence into single units from existing EDU-based discourse datasets. (2) For adjacent sentences, we determine their discourse relationship by identifying the lowest common parent node in the original EDU-based discourse tree. (3) We preserve these extracted relationships in a new tree structure with complete sentences as basic units. This conversion process allows us to train a parser $f_{\text{discourse}}$ that directly operates on sentence-level relationships while maintaining the essential discourse structure of texts.

Our core innovation lies in how we construct the document-level discourse tree through a two-phase iterative process that effectively preserves discourse relationships at multiple granularity levels:

**Phase 1. Paragraph-Level Tree Construction.** The initial phase processes each paragraph $p_i$ by transforming its sentence sequence $\boldsymbol{S_i} = \{s_{i,1}, s_{i,2}, ..., s_{i,m}\}$ into a local discourse tree:

$$T_i = f_{\text{discourse}}(\boldsymbol{S_i}), \tag{1}$$

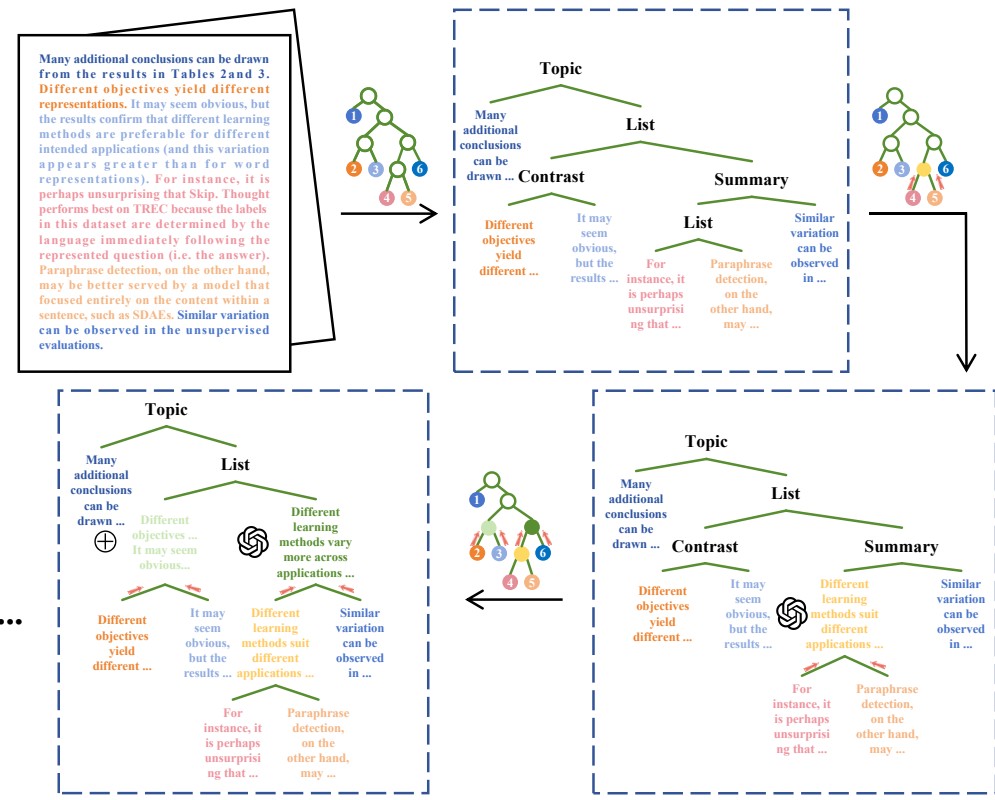

Figure 3: Illustration of bottom-up LLM enhancement in Phase 2 of discourse tree construction. Top: Input paragraph with its initial discourse tree structure. Center: Two-way processing strategy based on text length - using direct concatenation ($\bigoplus$) when combined length is below threshold $\tau$, and LLM-based summarization when above threshold. Bottom: Enhanced discourse tree with progressively generated semantic summaries following Equation 2, demonstrating the transformation from rhetorical relations to concrete semantic representations.

where each tree captures how sentences connect and build upon each other within its respective paragraph, reflecting both semantic relationships and logical flow between adjacent sentences.

**Phase 2. Document-Level Tree Construction.** The second phase obtains root text representation of paragraphs through bottom-up LLM enhancement. Since our discourse parser operates at sentence level, LLM enhancement helps simplify paragraph content into more manageable representations. For each internal node $v$ in the paragraph-level trees, we generate its text content according to:

$$t_v = \begin{cases} f_{\text{LLM}}(t_l, t_r), & \text{if } |t_l| + |t_r| \geqslant \tau \\ t_l \oplus t_r, & \text{otherwise} \end{cases} \quad (2)$$

where $t_l$ and $t_r$ are the text content of left and right children nodes. This adaptive summarization strategy reduces the complexity of multi-sentence content into more concise forms, making it more suitable for our sentence-level discourse parser. These root representations then serve as input for constructing the document-level discourse tree:

$$T_{\text{doc}} = f_{\text{discourse}}(\{t_{\text{root}}^1, t_{\text{root}}^2, ..., t_{\text{root}}^n\}). \quad (3)$$

As illustrated in Figure 3, this approach transforms rhetorical relations into concrete semantic summaries that effectively represent the content at different levels of the discourse hierarchy.

### 3.2 Node Representation

After obtaining both paragraph-level and document-level discourse trees from the previous construction process, we enhance the semantic representations of internal nodes through bottom-up LLM processing, integrate the enhanced trees into a unified hierarchical structure, and finally generate dense node embeddings to facilitate efficient semantic retrieval.

**Algorithm 1:** Hierarchical Evidence Retrieval and Selection

---

**Input:** Query $q$, discourse tree $T$, maximum evidence size $K$, leaf selection size $k$
**Output:** Evidence set $E$
```
/* Stage 1: Multi-level Evidence Retrieval                          */
```
$\mathbf{e_q} \leftarrow \text{Encoder}(q)$ ;                                                    // Transform query to embedding space
$\text{scores} \leftarrow \{\}, E \leftarrow \{\}$ ;
**for** *node* $v \in T$ **do**
  $\quad \mid \quad$ $\text{scores}[v] \leftarrow \text{cosine}(\mathbf{e_q}, \mathbf{e_v})$ ;                                // Compute similarity scores
**end**
$V_{\text{ranked}} \leftarrow \text{sort}(\text{scores}, \text{descending} = \text{True})$ ;
```
/* Stage 2: Structure-aware Evidence Selection                      */
```
$\text{used\_leaves} \leftarrow \{\}$ ;                                                    // Track selected leaves
**for** $v \in V_{\text{ranked}}$ **do**
  $\quad$ **if** $v$ *is leaf node* **then**
  $\quad\quad$ **if** $v \notin \text{used\_leaves}$ **then**
  $\quad\quad\quad$ $E \leftarrow E \cup \{v\}$ ;                                    // Direct selection of relevant leaves
  $\quad\quad\quad$ $\text{used\_leaves} \leftarrow \text{used\_leaves} \cup \{v\}$
  $\quad\quad$ **end**
  $\quad$ **else**
  $\quad\quad$ $L_v \leftarrow \text{get\_leaves}(v)$ ;                                // Get all leaves from subtree
  $\quad\quad$ $L'_v \leftarrow L_v \setminus \text{used\_leaves}$ ;                        // Remove already selected leaves
  $\quad\quad$ **if** $|L'_v| > 0$ **then**
  $\quad\quad\quad$ $L_{\text{topk}} \leftarrow \text{top\_k}(L'_v, \text{scores}, k)$ ;                // Leaf TopK Selection
  $\quad\quad\quad$ $E \leftarrow E \cup L_{\text{topk}}$ ;
  $\quad\quad\quad$ $\text{used\_leaves} \leftarrow \text{used\_leaves} \cup L_{\text{topk}}$
  $\quad\quad$ **end**
  $\quad$ **end**
  $\quad$ **if** $|E| \geq K$ **then**
  $\quad\quad$ **break**
  $\quad$ **end**
**end**
**return** $E$

---

**LLM Enhancement.** The document-level discourse tree $T_{\text{doc}}$ contains only discourse relationships at internal nodes. To obtain semantic representations for these nodes, we apply bottom-up LLM enhancement following Equation 2, similar to the paragraph-level process. This enhancement ensures that all nodes in the tree have meaningful semantic representations.

**Tree Integration.** After enhancing the document-level tree $T_{\text{doc}}$, we perform tree integration to combine it with all paragraph-level trees $T_i$ into a unified discourse structure. The integration process is straightforward: each leaf node in $T_{\text{doc}}$ is replaced with its corresponding paragraph-level discourse tree $T_i$, resulting in our final integrated tree $T_{\text{D}}$. This hierarchical structure effectively preserves both local sentence relationships within paragraphs and global discourse connections across the document.

**Node Encoding.** To enable efficient retrieval, we transform the discourse tree into an embedding tree structure using an off-the-shelf pre-trained sentence encoder. The resulting embedding tree preserves both the hierarchical structure and semantic information of the original discourse tree, making it suitable for downstream semantic matching tasks.

### 3.3 Hierarchical Evidence Retrieval and Selection

Our hierarchical evidence retrieval process introduces a novel structure-aware selection mechanism that leverages both local and global discourse information. Our approach combines:

- Direct selection of highly relevant leaf nodes
- Indirect selection through relevant internal nodes
- Automatic redundancy elimination via tree structure

Specifically, our algorithm first encodes the input query and computes similarity scores with all tree nodes. Then it processes nodes in order of decreasing relevance: leaf nodes are selected directly if highly relevant, while for internal nodes, we identify and select the most relevant unselected leaves from their subtrees. This process continues until we reach the desired evidence set size, ensuring both local relevance and structural coherence. The complete retrieval process is illustrated in Figure 2, demonstrating how evidence is gathered from different levels of the discourse tree. For implementation details, we refer readers to Algorithm 1.

# 4 Experiments

## 4.1 Experimental Setup

**Datasets and Evaluation Metrics.** We evaluate our approach on two challenging long document QA datasets: QASPER [10] for research paper understanding and QuALITY [27] for long-form reading comprehension. For QASPER, we use F1-Match for QA performance and token-level F1/Recall for retrieval evaluation. For QuALITY, we report accuracy as the primary metric. To ensure fair comparison, we fix the total length of retrieved context across all methods.

**Baselines.** We evaluate our approach against several strong retrieval baselines:

- Flatten-chunk splits articles into chunks of maximum 100 words while preserving sentence boundaries for semantic coherence.
- Flatten-sentence adopts sentence-level splitting and direct retrieval, providing a direct comparison baseline for our hierarchical approach.
- RAPTOR constructs a semantic tree through recursive embedding, clustering, and summarization, with retrieval performed on a collapsed tree structure following Sarthi et al. [31].
- Bisection shares our LLM-enhanced representations and hierarchical retrieval mechanism but uses a different tree construction method - recursively dividing sentence set into balanced binary subtrees. This baseline helps isolate benefits of our discourse-aware tree structure.

**Implementation details.** For discourse analysis, we construct our sentence-level RST training data based on RST-DT [6]. We then train our discourse parser following Yu et al. [38], adopting XLNet-base-cased as the pre-trained language model backbone. All summarization tasks, including both our approach (as defined in Equation 2) and RAPTOR baseline, utilize Llama3.1-8B-Instruct for consistency. We set the temperature parameter $\tau$ in Equation 2 to 0 for QASPER and 100 for QuALITY in our LLM enhancement process. For sentence encoder (Section §3.2), we experiment with both Sentence-BERT[1] and the latest embedding model of OpenAI[2] to evaluate the effectiveness of our method across different representation spaces. The Top-K parameter in leaf selection is set to 5 based on preliminary experiments (detailed in Appendix E.2). For generation models, we employ UnifiedQA-3B, GPT-4.1-mini and Deepseek-v3 to demonstrate the robustness of our retrieval improvements across different generation capabilities. More details are provided in Appendix D.

## 4.2 Main Results

**Generation Performance.** We conduct extensive experiments across different context lengths (200-400 words) and embedding models (SBERT and OpenAI). The results are shown in Table 1, comparing our approach with baselines on QASPER and QuALITY datasets.

Our experimental results demonstrate three key findings:

First, DISRetrieval consistently outperforms all baseline methods across different experimental settings. With 400-word contexts utilizing UnifiedQA-3B, it achieves substantial improvements over the flatten-sentence baseline (40.25% vs. 37.99% F1-Match on QASPER; 58.91% vs. 55.27% accuracy on QuALITY). These performance advantages remain robust and consistent across various generation models (UnifiedQA-3B, GPT-4.1-mini, and Deepseek-v3).

Second, compared to the tree-based RAPTOR baseline, DISRetrieval shows clear advantages. Using OpenAI embeddings with 300-word context, our method achieves 40.24% F1-Match versus RAPTOR of 39.96% on QASPER, demonstrating that incorporating linguistic discourse structure is more effective than purely relying on semantic-based clustering for document organization.

Finally, ablation studies with the Bisection variant highlight the importance of discourse structure in our approach. While Bisection improves over flatten-based methods through hierarchical modeling, it consistently underperforms the full DISRetrieval implementation across all settings. This empirically confirms that the integration of linguistic-informed discourse structure provides substantial benefits beyond simple hierarchical organization for long document understanding.

---

[1] multi-qa-mpnet-base-cos-v1, https://huggingface.co/sentence-transformers/multi-qa-mpnet-base-cos-v1
[2] OpenAI text-embedding-3-large

Table 1: Generation performance comparison of different methods under varying retrieved context lengths (200-400 words) and different embedding models (SBERT: s, OpenAI: o) across two datasets. Best results are **bolded**, runner-ups are underlined.

| Method | F1-Match (QASPER) / % | | | | | | Accuracy (QuALITY) / % | | | | | |
|---|---|---|---|---|---|---|---|---|---|---|---|---|
| | 200 (s) | 200 (o) | 300 (s) | 300 (o) | 400 (s) | 400 (o) | 200 (s) | 200 (o) | 300 (s) | 300 (o) | 400 (s) | 400 (o) |
| **UnifiedQA-3B** | | | | | | | | | | | | |
| flatten-chunk | 33.97 | 37.50 | 35.41 | 38.46 | 36.13 | 39.03 | 52.97 | 56.28 | 54.46 | 57.53 | 55.23 | 57.62 |
| flatten-sentence | 35.16 | 38.43 | 37.24 | 39.31 | 37.99 | 40.06 | 53.16 | 56.04 | 54.55 | 56.71 | 55.27 | 57.38 |
| RAPTOR | 33.57 | 37.46 | 34.95 | 39.96 | 37.00 | 39.53 | 53.60 | 55.99 | 54.75 | 57.00 | 55.51 | 58.53 |
| *Ours* | | | | | | | | | | | | |
| Bisection | 36.17 | 37.41 | 37.66 | 39.49 | 38.84 | 39.70 | 55.13 | 57.00 | 56.04 | 58.53 | 57.24 | 60.21 |
| DISRetrieval | **37.26** | **38.91** | **38.83** | **40.24** | **40.25** | **40.80** | **55.99** | **58.68** | **58.05** | **59.20** | **58.91** | **60.79** |
| **GPT-4.1-mini** | | | | | | | | | | | | |
| flatten-chunk | 37.37 | 41.13 | 41.38 | 43.34 | 42.72 | 44.78 | 60.16 | 65.77 | 63.66 | 69.27 | 67.69 | 71.05 |
| flatten-sentence | 39.82 | 43.08 | 41.84 | 45.28 | 42.44 | 45.78 | 60.12 | 64.43 | 63.09 | 66.68 | 65.24 | 69.94 |
| RAPTOR | 37.55 | 40.88 | 39.95 | 43.26 | 42.50 | 43.85 | 61.31 | 64.77 | 63.81 | 67.79 | 67.26 | 70.71 |
| *Ours* | | | | | | | | | | | | |
| Bisection | 40.98 | 43.12 | **42.74** | 44.54 | 43.49 | 45.69 | 63.71 | 67.88 | 65.68 | 70.71 | 67.93 | 72.00 |
| DISRetrieval | **41.04** | **43.47** | 42.58 | **45.47** | **44.52** | **45.97** | **63.71** | **68.79** | **66.87** | **71.43** | **69.51** | **73.20** |
| **Deepseek-v3** | | | | | | | | | | | | |
| flatten-chunk | 35.29 | 38.14 | 37.99 | 40.78 | 40.57 | 42.26 | 65.68 | 71.52 | 69.65 | 75.02 | 73.25 | 76.56 |
| flatten-sentence | 36.51 | 39.27 | 39.41 | 41.57 | 40.82 | 43.04 | 64.14 | 68.74 | 68.41 | 72.24 | 69.65 | 73.63 |
| RAPTOR | 34.66 | 37.84 | 37.30 | 40.50 | 39.77 | 42.17 | 65.53 | 69.13 | 68.65 | 72.24 | 71.19 | 75.22 |
| *Ours* | | | | | | | | | | | | |
| Bisection | 37.28 | 39.27 | 39.80 | 41.81 | 40.96 | **43.57** | 67.83 | 71.09 | 70.81 | 74.40 | 73.39 | 76.94 |
| DISRetrieval | **37.44** | **40.01** | **40.28** | **42.10** | **41.86** | 43.21 | **68.36** | **72.53** | **71.62** | **75.50** | **73.68** | **77.42** |

Table 2: Token-level F1 and Recall scores for retrieved contexts on the QASPER dataset across different methods using SBERT and OpenAI embedding model. Best results are **bolded** and runner-ups are underlined.

| Method | 200 (SBERT) | | 200 (OpenAI) | | 300 (SBERT) | | 300 (OpenAI) | | 400 (SBERT) | | 400 (OpenAI) | |
|---|---|---|---|---|---|---|---|---|---|---|---|---|
| | F1 / % | Recall / % | F1 / % | Recall / % | F1 / % | Recall / % | F1 / % | Recall / % | F1 / % | Recall / % | F1 / % | Recall / % |
| flatten-chunk | 26.13 | 56.05 | 29.17 | 62.16 | 23.10 | 63.92 | 25.12 | 69.04 | 20.38 | 68.75 | 21.91 | 73.38 |
| flatten-sentence | 26.15 | 57.80 | 28.25 | 62.78 | 22.68 | 64.63 | 24.04 | 68.43 | 19.98 | 69.02 | 21.06 | 72.42 |
| RAPTOR | 24.57 | 52.42 | 27.18 | 58.49 | 21.71 | 60.00 | 23.57 | 65.63 | 19.27 | 65.11 | 20.64 | 70.04 |
| *Ours* | | | | | | | | | | | | |
| Bisection | 27.63 | 59.82 | 29.29 | 63.11 | 23.83 | 66.69 | 25.16 | 69.94 | 21.10 | 71.52 | 21.98 | 74.18 |
| DISRetrieval | **28.45** | **61.28** | **30.29** | **64.92** | **24.67** | **68.25** | **26.05** | **71.62** | **21.70** | **73.09** | **22.63** | **75.77** |

**Retrieval Performance.** We evaluate the retrieval effectiveness of DISRetrieval on the QASPER dataset using token-level F1 and Recall metrics. The results in Table 2 demonstrate three key findings:

First, DISRetrieval consistently outperforms baselines across all settings, achieving the highest F1 and Recall scores with both SBERT and OpenAI embeddings. This demonstrates that our discourse-aware context modeling effectively captures the semantic relationships within documents.

Second, for longer contexts (300-400 words), while all methods show some F1 score degradation, DISRetrieval maintains superior performance, particularly in Recall metrics. This robust performance on longer contexts validates the capability of our method in handling complex document structures through discourse-guided retrieval.

The ablation with Bisection shows that while hierarchical organization helps, the full discourse-aware approach provides additional benefits. Additionally, OpenAI embeddings consistently outperform SBERT across all settings, suggesting that strong semantic representations are fundamental to the effectiveness of discourse-aware retrieval.

## 4.3 Discussions

**RQ1: Is the hierarchical retrieval strategy effective? Fig 4** In Section 3.3, we propose a carefully designed hierarchical retrieval strategy composed of multiple sub-steps and associated methods. To assess the contribution of each component, we conduct ablation experiments on the QuALITY dataset to validate the effectiveness of each component in our hierarchical retrieval strategy. As shown in Figure 4, we compare five variants: leaf-only baseline, summary-based retrieval, all filtered-leaves, Top-K with ranking order, and our final Top-K with original order.

The results reveal three key insights: (1) First, using summaries of intermediate nodes performs worse than the leaf baseline, indicating that preserving original text details is crucial. (2) Second, while

using all filtered leaves shows slight improvements, it is less effective than selective Top-K methods due to noise from irrelevant content. (3) Most importantly, our Top-K origin method, which preserves the natural document flow, consistently achieves the best performance across all settings. This advantage becomes more pronounced with longer contexts and better embeddings, demonstrating that maintaining document structure is essential for effective retrieval.

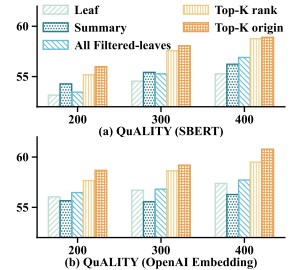

Figure 4: Ablation results of different variants.

Table 3: Effect of different LLMs for node text enhancement on QASPER dataset retrieval performance (token-level F1 and Recall) across varying context lengths (200, 300, and 400 words).

| | 200 | | 300 | | 400 | |
|---|---|---|---|---|---|---|
| | F1 / % | Recall / % | F1 / % | Recall / % | F1 / % | Recall / % |
| Llama-3.1-8B | 28.45 | 61.28 | 24.67 | 68.25 | 21.70 | 73.09 |
| Qwen2.5-7B | 28.32 | 61.09 | 24.75 | 68.11 | 21.69 | 72.53 |
| Mistral-7B | 28.17 | 60.86 | 24.65 | 68.41 | 21.65 | 72.86 |
| GPT-4o-mini | 28.27 | 61.13 | 24.66 | 68.31 | 21.68 | 72.84 |
| Deepseek-v3 | 28.89 | 61.91 | 25.02 | 68.57 | 21.82 | 72.91 |

**RQ2: Does the scale of LLMs affect the quality of the discourse-aware tree structures in node text enhancement? Tab 3** We investigate whether model scale affects the quality of discourse-aware tree construction by comparing different LLMs. As shown in Table 3, smaller models like Llama-3.1-8B, Qwen2.5-7B, and Mistral-7B achieve comparable performance to larger models, with differences under 0.5%. Notably, Llama-3.1-8B achieves the best recall at 400-word context length. These results indicate that the effectiveness of our discourse-aware retrieval does not heavily depend on LLM scale. This finding is particularly valuable as it enables flexible model selection without requiring extremely large models, making our approach more practical and cost-effective.

**RQ3: Does the capability of the discourse parser have a significant impact? Fig 5** We examine how discourse parser capabilities affect downstream performance by training parsers on varying amounts of data (0-100%). As shown in Figure 5, both retrieval recall and answer F1 scores improve consistently as parser training data increases across all context lengths. These results confirm that parser quality directly impacts the effectiveness of our discourse-aware tree structure. This suggests that improving discourse parsing capabilities is a promising direction for enhancing our approach's performance.

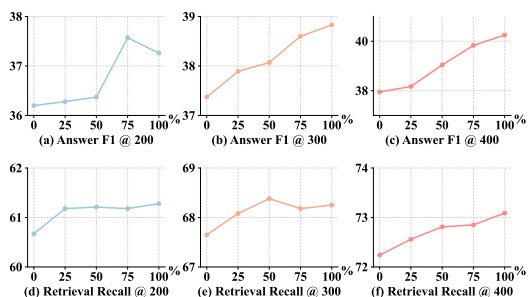

Figure 5: Impact of discourse parser capability on subsequent retrieval and question answering performance. Parsers used for comparative evaluation are trained on varying data scales, ranging from 0% to 100%.

## 5 Conclusion

In this paper, we presented DISRetrieval, a discourse-aware hierarchical retrieval framework that advances long document understanding through three key innovations: (1) a multi-granularity adaptive discourse framework based on RST, (2) an LLM-enhanced node representation technique, and (3) a structure-aware evidence selection mechanism. Our evaluation on QASPER and QuALITY benchmarks demonstrates significant improvements over existing methods. The consistent performance advantages across varying document lengths and embedding models validate the effectiveness of our discourse-aware approach. Through detailed ablation studies, we identified that the combination of discourse-aware tree construction and hierarchical retrieval mechanism contributes most significantly to the performance gains, highlighting the crucial role of linguistic structure in document understanding. Looking forward, we envision three promising directions: (1) extending our framework to cross-lingual document retrieval, (2) incorporating more sophisticated discourse theories, and (3) exploring dynamic discourse structure adaptation for different domains. Our work establishes a new paradigm for leveraging linguistic structures in information retrieval systems, paving the way for more effective long-form content understanding.

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

## A    Limitation and Future Work

We here discuss the limitations and future work of DISRetrieval.

First, while our discourse-aware tree construction method's performance is inherently bounded by the quality of RST parsing, our current approach demonstrates significant improvements even with a straightforward adaptation from EDU-level to sentence-level RST structures. This success validates the effectiveness of incorporating discourse information in document retrieval. However, we believe there is still substantial room for enhancement. In future work, we plan to explore more sophisticated discourse analysis techniques and develop specialized tree construction methods that can better capture the hierarchical relationships in long documents.

Second, although we have considered computational efficiency by implementing an adaptive summarization strategy with a length threshold $\tau$ to determine when LLM processing is necessary, this approach is relatively simple. While effective, it may not always achieve an optimal balance between computational cost and representation quality. Future work will investigate more sophisticated methods for managing computational resources, such as dynamic thresholding based on content complexity, hierarchical position-aware summarization, and more efficient LLM architectures specifically designed for tree-structured document processing.

Third, while our evaluation on QASPER and QuALITY datasets demonstrates significant improvements in traditional retrieval metrics, the assessment of long document understanding could benefit from more diverse evaluation criteria. Current metrics may not fully capture the nuanced aspects of discourse-aware retrieval, such as structural coherence preservation and hierarchical information flow. We plan to develop more comprehensive evaluation frameworks that consider multiple dimensions of retrieval quality, including discourse coherence, semantic relevance, and structural preservation at different granularity levels.

These limitations point to promising directions for future research while reinforcing the solid foundation and significant contributions of our current work. We believe addressing these aspects will further advance the field of discourse-aware document retrieval and enhance its practical applications.

## B    Potential Impact

The current approaches to long document retrieval often struggle with maintaining semantic coherence and structural integrity, primarily due to the lack of discourse-aware document modeling. Our work, DISRetrieval, addresses this fundamental challenge by introducing a novel framework that leverages discourse structure for enhanced document understanding and retrieval.

The impact of our work extends beyond mere performance improvements in retrieval metrics. By demonstrating the effectiveness of discourse-aware hierarchical modeling, we provide the research community with a new paradigm for approaching long document understanding tasks. Our framework shows that incorporating discourse structure not only enhances retrieval accuracy but also preserves the natural organization of information, which is crucial for human comprehension.

Moreover, our LLM-enhanced node representation technique introduces a novel way of combining traditional discourse analysis with modern LLMs. This hybrid approach could inspire similar innovations in other NLP tasks where maintaining structural relationships while leveraging powerful language models is essential. The adaptive summarization strategy we proposed, although currently implemented with a simple threshold mechanism, opens up new research directions for efficient document processing with LLMs.

Following this line of research, similar discourse-aware approaches could be applied to various challenging tasks in document understanding, such as long-form question answering, document summarization, and information extraction. Our work provides the community with a new perspective on handling long documents by emphasizing the importance of discourse structure, potentially influencing how future systems approach the challenge of processing and understanding lengthy textual content.

Furthermore, the success of DISRetrieval in bridging the gap between document structure and semantic understanding suggests promising applications in real-world scenarios, particularly in

domains where maintaining the logical flow of information is crucial, such as legal document analysis, scientific literature review, and educational content processing.

# C Extension of Technical Details

In this section, we introduce the specific details of our method which we cannot present in the main article due to the space limit.

## C.1 Discourse Parser Details

The discourse parser architecture builds upon a transition-based system that constructs discourse trees through a series of well-defined actions. This system is particularly effective for capturing both local and global discourse relationships while maintaining computational efficiency.

The transition system maintains two primary data structures: a stack $\sigma$ that holds partially built trees, and a queue $\beta$ that contains unprocessed sentences. This design follows the intuition that discourse relationships often exist between adjacent text spans and can be built incrementally from bottom to top.

The system builds the tree step by step using three basic operations:

1. A "shift" action moves a sentence from the queue to the stack when we need new content to process.

2. A "reduce" action combines two adjacent subtrees on top of the stack into a new subtree by identifying their discourse relationship.

3. A "pop root" action concludes the process when we have successfully built a complete tree.

Each state of the system is represented as $c = (\sigma, \beta)$, starting from $c_0 = ([\,], S_i)$ with all sentences in the queue, and ending at $c_f = ([T_i], [\,])$ with a complete discourse tree $T_i$.

The transition system follows a deterministic process guided by the neural scoring model:

1. Initialize $\sigma = [\,]$ and $\beta = S_i$.

2. While $\beta$ is not empty or $|\sigma| > 1$: (a) If $|\sigma| < 2$ and $\beta$ is not empty, perform a "shift" action to move the next sentence from $\beta$ to $\sigma$; (b) Else if $\beta$ is empty, perform a "reduce" action to combine the top two subtrees in $\sigma$; (c) Else, use the neural scoring model to decide between a "shift" or "reduce" action based on the current state of $\sigma$ and $\beta$.

3. Return the single tree $T_i$ remaining on the stack $\sigma$.

The scoring model considers the three topmost subtrees on the stack $(s_1, s_2, s_3)$ and the next sentence in the queue $q_1$. This design is motivated by several factors:

1. $s_1$ and $s_2$ are the immediate candidates for the next potential "reduce" action.

2. $s_3$ provides crucial context about the recently built structure.

3. $q_1$ helps determine if we should introduce new content via a "shift" action.

For each tree node $v$, we compute its representation $h_v$ recursively:

$$\mathbf{h}_v = \begin{cases} \text{PLM}(s_i), & \text{if } v \text{ is a sentence} \\ \frac{1}{|C(v)|} \sum_{u \in C(v)} \mathbf{h}_u, & \text{if } v \text{ is a relationship node} \end{cases} \tag{4}$$

where $C(v)$ denotes the set of child nodes of $v$, and $\text{PLM}(\cdot)$ is a pre-trained language model that encodes the semantic meaning of individual sentences. The action scores are then computed as:

$$\mathbf{y}(a) = \mathbf{W}(\mathbf{h}_{s_1} \oplus \mathbf{h}_{s_2} \oplus \mathbf{h}_{s_3} \oplus \mathbf{h}_{q_1}) + \mathbf{b}, \tag{5}$$

where $\oplus$ concatenates the representations to capture their interactions, and $\mathbf{W}$ and $\mathbf{b}$ are learnable parameters. The probability of taking action $a$ is computed using a softmax function over the action scores:

$$p(a|c) = \frac{\exp(\mathbf{y}(a))}{\sum_{a' \in A} \exp(\mathbf{y}(a'))}, \tag{6}$$

---

**Algorithm 2:** DISRetrieval: Discourse Structure-based Long Document Retrieval

---

**Input:** Document $D$ containing sentences $S_i$, paragraphs $P_i$; Query $q$
**Output:** Retrieved evidence segments $E$
```
/* Stage 1:  Discourse-Aware Tree Construction                              */
```
**for** *each paragraph* $P_i \in D$ **do**
    Initialize stack $\sigma$ and sentence queue $\beta \leftarrow S_i$;
    **while** $\beta$ *not empty OR* $|\sigma| > 1$ **do**
```
        /* RST parsing operations                                           */
```
        $a_t \leftarrow$ Select action from {shift, reduce, pop_root};         // Determine next parsing action
        **switch** $a_t$ **do**
            **case** *shift* **do**
                Move next sentence from $\beta$ to $\sigma$;         // Add new sentence to stack
            **end**
            **case** *reduce* **do**
                $s_2, s_1 \leftarrow$ Pop top two elements from $\sigma$;
                Determine discourse relation $r$ between $s_1$ and $s_2$;         // Using RST relations
                Create new node with relation $r$;
                Push new node to $\sigma$;
            **end**
            **case** *pop_root* **do**
                Final tree node $\leftarrow$ Pop from $\sigma$;         // Complete the tree
            **end**
        **end**
    **end**
    $T_i \leftarrow$ Resulting discourse tree;
```
    /* LLM Enhancement for Paragraph Tree                                   */
```
    **for** *each non-leaf node* $v \in T_i$ *(bottom-up order)* **do**
        $t_l, t_r \leftarrow$ Get content from left and right children;
        **if** $|t_l| + |t_r| \geq \tau$ **then**
            $t_v \leftarrow f_{\text{LLM}}(t_l, t_r)$;         // Generate concise summary using LLM
        **else**
            $t_v \leftarrow t_l \oplus t_r$;         // Direct concatenation for short text
        **end**
    **end**
    Store root text representation $t_{\text{root}}^i$ for $T_i$;
**end**
Initialize stack $\sigma$ and queue $\beta \leftarrow \{t_{\text{root}}^1, ..., t_{\text{root}}^n\}$;
**while** $\beta$ *not empty OR* $|\sigma| > 1$ **do**
```
    /* Similar RST parsing at document level                                */
```
    Apply RST parsing operations as in Phase 1;
    Build document tree $T_{\text{doc}}$;
**end**
```
/* Stage 2:  Node Representation                                            */
```
**for** *each non-leaf node* $v \in T_{doc}$ *(bottom-up order)* **do**
    $t_l, t_r \leftarrow$ Get content from left and right children;
    **if** $|t_l| + |t_r| \geq \tau$ **then**
        $t_v \leftarrow f_{\text{LLM}}(t_l, t_r)$;         // LLM-based enhancement for document-level nodes
    **else**
        $t_v \leftarrow t_l \oplus t_r$;         // Direct concatenation for short text
    **end**
**end**
$T_D \leftarrow$ Replace leaf nodes in $T_{\text{doc}}$ with corresponding $T_i$;         // Integrate trees into unified structure
**for** *each node* $v \in T_D$ **do**
    $e_v \leftarrow$ Encoder$(t_v)$;         // Generate dense vector representations
**end**
```
/* Stage 3:  Hierarchical Evidence Retrieval and Selection                  */
```
$e_q \leftarrow$ Encoder$(q)$;         // Transform query to embedding space
scores $\leftarrow \{\}, E \leftarrow \{\}$;
**for** *node* $v \in T_D$ **do**
    scores$[v] \leftarrow$ cosine$(e_q, e_v)$;         // Compute relevance scores
**end**
Apply Algorithm 1 for evidence selection;
**return** $E$

---

where $A$ is the set of valid actions at state $c$. We train the model using supervised learning with gold-standard discourse trees. The objective function combines cross-entropy loss for action prediction with L2 regularization:

$$\mathcal{L}(\theta) = -\log p(a^*|c) + \frac{\lambda\|\theta\|_2}{2}, \tag{7}$$

where $a^*$ is the correct action derived from gold-standard trees. During inference, we greedily select the highest-scoring action at each step, effectively building the tree in a bottom-up manner while maintaining the discourse relationships between text spans.

## C.2  Iterative Tree Construction Process

The complete algorithm of our DISRetrieval is presented in Algorithm 2. Below we detail the iterative tree construction process, which corresponds to Stage 1 of the algorithm. The construction of discourse trees for long documents presents unique challenges in balancing computational efficiency with structural integrity. We propose an iterative construction strategy that addresses these challenges through a hierarchical, phase-wise approach. This method effectively manages computational resources while preserving discourse relationships at multiple granularity levels. Our iterative process consists of three distinct phases, each designed to handle specific aspects of the tree construction:

**Phase 1: Paragraph-level Tree Construction.** The first phase focuses on building local discourse trees for individual paragraphs. For each paragraph $p_i$ containing sentences $\boldsymbol{S_i} = \{s_{i,1}, s_{i,2}, ..., s_{i,m}\}$, we construct a local discourse tree $T_i$ using our transition-based parsing system. This phase is particularly efficient as it:

1. Initializes each paragraph's parsing state with an empty stack and sentence queue: $c_0 = ([\,], S_i)$.
2. Processes paragraphs independently, enabling parallel computation.
3. Applies transition actions iteratively until a complete tree is formed.
4. Stores both the resulting paragraph-level tree $T_i$ and its root representation $\mathbf{h}_{T_i}$.

**Phase 2: Document-level Tree Construction.** The second phase focuses on capturing document-level discourse structure. After obtaining all paragraph-level trees $T_1, T_2, ..., T_n$, we:

1. For each paragraph-level tree $T_i$, apply bottom-up LLM-enhanced summarization:
   - For each non-leaf node $v$ with children $c_l$ and $c_r$:

$$t_v = \begin{cases} f_{\text{LLM}}(t_l, t_r), & \text{if } |t_l| + |t_r| \geqslant \tau \\ t_l \oplus t_r, & \text{otherwise} \end{cases} \tag{8}$$

   where $t_l$ and $t_r$ are the textual content of child nodes
   - Continue until reaching root node to obtain semantic unit $u_i$
2. Form the semantic units set $U = \{u_1, u_2, ..., u_n\}$ from root representations
3. Apply the discourse parser to these units to construct a document-level tree $T_{\text{doc}}$ using the same transition-based parsing system:

$$T_{\text{doc}} = f_{\text{discourse}}(U) \tag{9}$$

4. Apply bottom-up LLM-enhanced summarization to $T_{\text{doc}}$:
   - For each non-leaf node $v \in T_{\text{doc}}$ with children $c_l$ and $c_r$:

$$t_v = \begin{cases} f_{\text{LLM}}(t_l \oplus t_r), & \text{if } |t_l \oplus t_r| \geqslant \tau \\ t_l \oplus t_r, & \text{otherwise} \end{cases} \tag{10}$$

   - Process nodes level by level from bottom to top until reaching the root of $T_{\text{doc}}$

This step effectively captures the high-level discourse relationships between paragraphs while maintaining computational efficiency by working with LLM-enhanced condensed representations at both paragraph and document levels.

**Tree Integration.** The final phase integrates the local and global structures into a unified discourse tree. This integration is accomplished through a careful replacement process:

$$T_{\text{D}} = \text{Replace}(T_{\text{doc}}, \{(u_i, T_i) | i \in [1, n]\}) \tag{11}$$

where $\text{Replace}(\cdot)$ replaces each unit $u_i$ in the document-level tree with its corresponding paragraph-level tree $T_i$. This integration phase carefully preserves both local and global discourse relationships by maintaining the internal structure of paragraph-level trees while retaining the document-level relationships established in Phase 2, ultimately creating a seamless hierarchical structure that spans the entire document and effectively captures discourse relationships at all levels of granularity.

Table 4: Detailed infomation of the datasets used in our experiments.

| Dataset | Used Set | Question Num | Document Num | Avg. Document Length (words) |
|---------|----------|--------------|--------------|------------------------------|
| QASPER | test | 1456 | 416 | 4170 |
| QuALITY | dev | 2086 | 115 | 5022 |

The resulting tree structure effectively captures discourse relationships at multiple levels of granularity, from sentence-level connections within paragraphs to broader document-level organizational patterns. This approach not only ensures computational efficiency through its phase-wise processing but also maintains the integrity of discourse relationships throughout the document hierarchy.

# D Detailed Experiment Settings

## D.1 Dataset Specifications

The QASPER dataset is constructed from NLP research papers, containing questions that require deep understanding of technical content. All answerable questions are annotated with multiple reference answers to account for different valid expressions of the same information. The ground-truth evidence spans are carefully annotated by domain experts to ensure the reliability of retrieval evaluation.

The QuALITY dataset consists of long passages primarily drawn from fiction stories and magazine articles, with an average length significantly longer than typical QA datasets. Each question is accompanied by multiple choice options that test comprehensive understanding of the passage. We utilize the validation set with publicly available labels for our experiments to enable thorough analysis and comparison. Notably, to enable more extensive experimentation, we evaluated on the validation set with publicly available labels rather than the submission-required test set. More detailed information is presented in Table 4.

## D.2 Model Specifications

For semantic embeddings, the Sentence-BERT model (multi-qa-mpnet-base-cos-v1) is based on the MPNet architecture with 768-dimensional representations, specifically optimized for question-answering tasks. The OpenAI embedding model (text-embedding-3-large) represents their latest advancement in semantic representation capabilities.

## D.3 Implementation Details

In the discourse analysis process, we preserve sentence boundaries throughout all splitting operations to maintain semantic coherence. For the RAPTOR baseline implementation, we follow the original paper's collapsed tree approach where all nodes are considered simultaneously during retrieval. The tree construction process in our Bisection baseline ensures a nearly balanced binary structure through recursive division of the sentence set.

## D.4 Evaluation Protocol

For QASPER's retrieval evaluation, we compute token-level F1 and Recall scores by comparing the retrieved context against the annotated ground truth evidence spans. This provides a fine-grained assessment of retrieval quality beyond simple overlap metrics. For QuALITY, the accuracy metric reflects the proportion of correctly answered multiple-choice questions, directly measuring the impact of retrieval quality on downstream QA performance.

## D.5 Experiments Compute Resources

For the training of the sentence-level Discourse Parser, we used 1 NVIDIA A100-40G GPU. All other experiments incluing document discourse parsing, LLM summarization and node embedding, as well as the retrieval and generation processes, are conducted using 4 NVIDIA A800-80G GPUs.

 ## D.6 Prompts

We provides the specific prompts used in our approach to ensure reproducibility.

In the prompt templates: (1) fixed prompts are displayed in black. (2) Input text is highlighted in deep red. (3) The retrieved context is colored in purple. (4) The output generated by the LLM is presented in green.

---

**Prompt for Intermediate Node Text Summerization**

**System**: You are a helpful assistant.

**User**: Write a summary of the given sentences, keeps as more key information as possible. Only give the summary without other text. Make sure that the summary no more than 200 words.
Given text: {left child node text} {input cild node text}

**LLM**: Summerization result.

---

**Prompt for Question Answering on QASPER Dataset**

**System**: You are a helpful assistant.

**User**: Using the folling information: {context}. Answer the following question in less than 5-7 words, if possible. For yes or no question, only return 'yes' or 'no'.
question: {question}

**LLM**: Question Answering result.

---

**Prompt for Question Answering on QuALITY Dataset**

**System**: You are a helpful assistant.

**User**: Given context: {context}.
Answer the following multiplie-choice question:{question}

**LLM**: The correct answer is (A). The context provided...

---

# E    Extended Experiment Results and Analyses

## E.1    Analysis of Multi-granularity Adaptive Retrieval

We conducted a thorough statistical analysis to evaluate the multi-granularity adaptive retrieval capability of our method. For each query, we retrieved the Top-20 nodes from the full discourse-aware tree, and analyzed the characteristics of retrieved intermediate nodes across datasets.

**Retrieved Node Characteristics Across Datasets.**    Table 5 reveals significant differences in the retrieved intermediate nodes between datasets. QuALITY exhibits substantially higher values for intermediate node metrics compared to QASPER: the average depth of retrieved intermediate nodes is 2.6 times greater with SBERT embeddings (13.78 vs. 5.29) and 3.1 times greater with OpenAI embeddings (17.09 vs. 5.53). Similarly, the average number of leaf nodes that each retrieved intermediate node maps to is 5.4 times higher with SBERT (67.67 vs. 12.64) and 6.6 times higher with OpenAI (88.67 vs. 13.37). The percentage of intermediate nodes among Top-20 retrieved results is also higher for QuALITY across both embedding methods (73.93% vs. 54.15% with SBERT; 88.53% vs. 65.13% with OpenAI). These differences exist despite QASPER having a 50% longer average sentence length (22.32 vs. 14.86 words).

Table 5: Statistical analysis of retrieved intermediate node characteristics across QuALITY and QASPER datasets. The table compares various key metrics: *Avg. Sentence Length* means the average sentence length of all documents; *Avg. Mid Node Depth* is the average depth of retrieved intermediate nodes; *Avg. Leaf Num* represents the average number of leaf nodes that each retrieved intermediate node maps to; *Avg. Mid Node Percentage* is the average percentage of intermediate nodes among the Top-20 retrieved nodes.

| Dataset | Avg. Sentence Length | Avg. Mid Node Depth | | Avg. Leaf Num | | Avg. Mid Node Percentage / % | |
| --- | --- | --- | --- | --- | --- | --- | --- |
| | | SBERT | OpenAI | SBERT | OpenAI | SBERT | OpenAI |
| QASPER | 22.32 | 5.29 | 5.53 | 12.64 | 13.37 | 54.15 | 65.13 |
| QuALITY | 14.86 | 13.78 | 17.09 | 67.67 | 88.67 | 73.93 | 88.53 |

**Distributional Analysis.** Figure 6 further illustrates these distinctions through distributional analysis. The sentence length distribution (Fig. 6a) shows that QuALITY is heavily skewed toward shorter sentences (5-15 words), while QASPER has a broader, more even distribution extending to 40+ words. The node depth distributions (Fig. 6b,c) demonstrate that retrieved intermediate nodes from QuALITY frequently reach depths of 10-15, whereas those from QASPER are predominantly shallower (depths below 10), consistent across both embedding methods.

**Retrieval Strategy Adaptation.** These patterns reveal how our retrieval approach adapts to different document structures. For narrative fiction in QuALITY with abundant dialogue and shorter sentences, the retrieval system favors higher-level intermediate nodes that aggregate related content, as evidenced by the higher percentage of intermediate nodes and greater node depths. This suggests that hierarchical composition is particularly beneficial for documents where meaning is distributed across multiple short sentences. In contrast, for research papers in QASPER with longer, more self-contained sentences, the retrieval system relies less on deep hierarchical structures, as shown by the shallower node depths and lower percentage of intermediate nodes. This indicates that relevant information in scientific documents can often be retrieved effectively with less compositional processing.

These findings empirically validate that our approach effectively adapts to diverse document structures and information needs. Our discourse-aware tree construction naturally induces appropriate segmentations based on document characteristics rather than arbitrary chunking, enabling multi-level text composition that overcomes the limitations of fixed-granularity methods. Through these mechanisms, DISRetrieval provides a multi-granularity retrieval framework that adaptively selects appropriate granularity levels across different document types, accommodating their inherent structural variations.

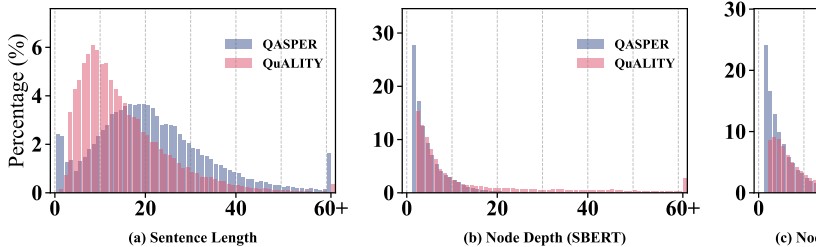 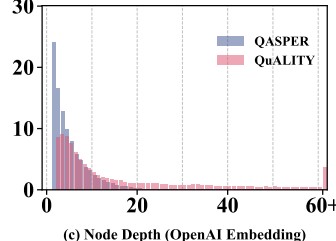

Figure 6: Comparative analysis of distribution difference of two datasets. Figure (a) shows the difference on Sentence Length, while (b) and (c) demonstrate the distribution of retrieved intermediate node depth across different embedding models.

### E.2 Ablation study on different Top-K settings

We conduct comprehensive experiments to investigate the impact of varying the value of $K$ from 1 to 20, with results presented in Figure 7.

Our analysis reveals several noteworthy patterns:

**Performance Trends.** Across all configurations, performance initially improves as $K$ increases, typically peaking around $K = 5$ (marked by the vertical dashed line), after which it either plateaus,

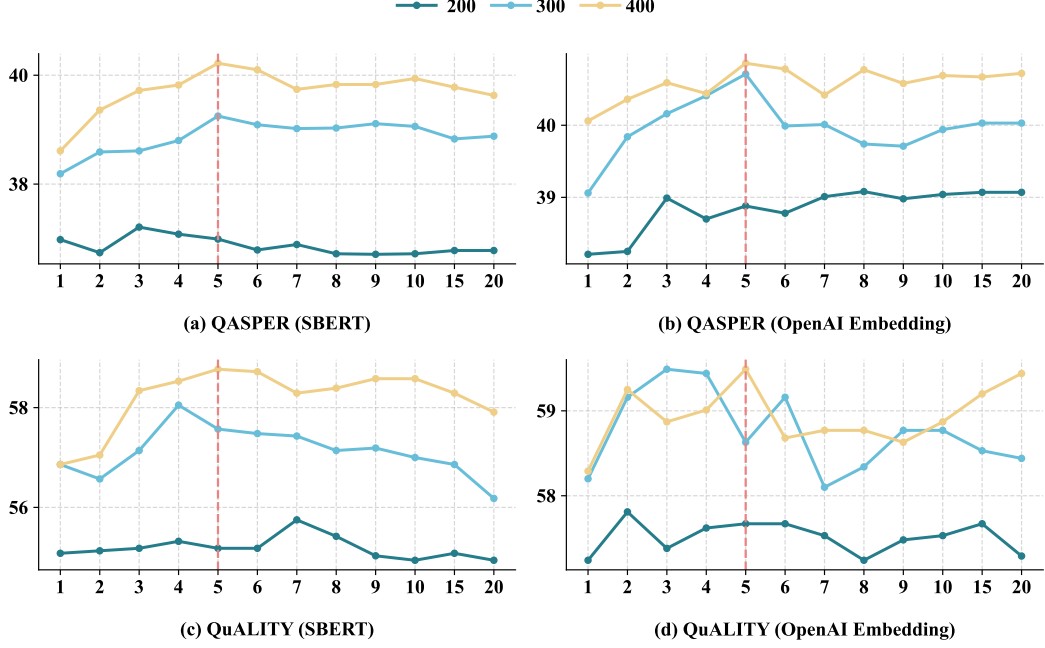

Figure 7: Ablation results different values of $K$. The horizontal axis represents different choices of $K$, and the vertical axis indicates generation performance (F1-match for QASPER and accuracy for QuALITY). All question answering tasks are conducted on the UnifiedQA-3B model.

gradually declines, or exhibits minor fluctuations. This consistent pattern suggests an optimal balance point where sufficient context is provided without introducing excessive noise.

**Dataset-Specific Behaviors.** QuALITY demonstrates more pronounced performance variations with changing $K$ values compared to QASPER, with performance differences of up to 2 percentage points between optimal and suboptimal settings. This higher sensitivity likely reflects QuALITY's more complex narrative structure, where precise evidence selection is particularly crucial.

**Context Length Impact.** Longer context windows (300 and 400 words) consistently outperform shorter ones (200 words) across all $K$ values. Notably, the performance advantage of longer contexts is most significant when $K$ is small. This suggests that when the system retrieves fewer evidence segments, the comprehensiveness of each individual segment becomes critical, as the model must extract all necessary information from a limited number of passages.

Based on these observations, we adopt $K = 5$ as our optimal setting for all main experiments, as it consistently delivers strong performance across datasets and embedding methods while maintaining computational efficiency.

## E.3 Detailed Results of different answer types on QASPER Dataset

The question-answering task on the QASPER dataset includes four types of answers: Abstractive, Yes / No, Extractive and Answerable / Unanswerable.

- Abstractive: This type of answers use abstract and summarized content rather than the original text from the document. They often involve rephrasing or synthesizing content to provide a concise and coherent response.

- Yes / No: These answers are straightforward and binary, simply *Yes* or *No* to the question.

- Extractive: These answers are directly taken from the original text. The goal is to pinpoint and extract the specific segment of text that directly answers the question.

- Answerable / Unanswerable: This type categorizes whether the question can be answered based on the information provided in the text. These type of answers correspond to questions that the original documents do not contain sufficient supporting information, and the model is expected to directly reply with "unanswerable."

We present the detailed results for each type of answer in Table 6. Notably, both GPT-4.1-mini and Deepseek-v3 achieve zero scores on the Answerable / Unanswerable type due to their inability to meet the specific answer word requirement.

Table 6: Question answering performance across different answer types on QASPER dataset. *Abs*, *Bool*, *Ext*, and *None* represent Abstractive, Yes / No, Extractive and Answerable / Unanswerable answers respectively. *Final* denotes the average F1 score across different answer types.

| Method | 200 | | | | | 300 | | | | | 400 | | | | |
|---|---|---|---|---|---|---|---|---|---|---|---|---|---|---|---|
| | Abs | Bool | Ext | None | Final | Abs | Bool | Ext | None | Final | Abs | Bool | Ext | None | Final |
| **UnifiedQA-3B + SBERT** | | | | | | | | | | | | | | | |
| flatten-chunk | 16.37 | 73.30 | 32.68 | 18.85 | 33.97 | 16.26 | 75.45 | 35.16 | 18.18 | 35.41 | 15.73 | 75.91 | 36.82 | 15.83 | 36.13 |
| flatten-sentence | 16.69 | 75.34 | 34.46 | 17.74 | 35.16 | 16.69 | 76.68 | 38.20 | 15.13 | 37.24 | 17.61 | 77.48 | 39.6 | 11.67 | 37.99 |
| raptor | 16.26 | 71.95 | 32.59 | 18.25 | 33.57 | 16.32 | 72.48 | 35.19 | 17.36 | 34.95 | 16.97 | 77.27 | 37.28 | 18.03 | 37.00 |
| *Ours* | | | | | | | | | | | | | | | |
| Bisection | 16.13 | 77.38 | 36.53 | 15.57 | 36.17 | 16.57 | 76.92 | 39.49 | 12.71 | 37.66 | 17.03 | 75.78 | 41.81 | 10.34 | 38.84 |
| DISRetrieval | 17.30 | 77.48 | 37.63 | 15.57 | 37.26 | 18.55 | 75.68 | 41.02 | 11.57 | 38.83 | 18.50 | 77.48 | 42.96 | 13.33 | 40.25 |
| **UnifiedQA-3B + OpenAI Embedding** | | | | | | | | | | | | | | | |
| flatten-chunk | 16.33 | 75.45 | 38.77 | 18.03 | 37.50 | 17.39 | 76.13 | 39.95 | 17.24 | 38.46 | 17.68 | 77.27 | 41.06 | 17.36 | 39.03 |
| flatten-sentence | 17.33 | 75.45 | 40.86 | 13.56 | 38.43 | 17.53 | 76.71 | 42.11 | 13.68 | 39.31 | 18.69 | 77.06 | 43.16 | 10.43 | 40.06 |
| raptor | 15.68 | 74.21 | 39.51 | 15.65 | 37.46 | 16.91 | 77.03 | 42.72 | 15.18 | 39.96 | 16.47 | 76.92 | 42.39 | 13.16 | 39.53 |
| *Ours* | | | | | | | | | | | | | | | |
| Bisection | 17.27 | 73.76 | 39.70 | 11.11 | 37.41 | 18.03 | 76.47 | 42.77 | 8.33 | 39.49 | 18.21 | 77.73 | 43.22 | 5.88 | 39.70 |
| DISRetrieval | 18.13 | 77.73 | 41.5 | 9.24 | 38.91 | 19.67 | 77.93 | 43.12 | 7.69 | 40.24 | 20.02 | 77.38 | 44.57 | 5.13 | 40.80 |
| **GPT-4.1-mini + SBERT** | | | | | | | | | | | | | | | |
| flatten-chunk | 21.16 | 78.09 | 38.48 | – | 37.37 | 23.85 | 84.55 | 42.78 | – | 41.38 | 24.90 | 84.17 | 44.76 | – | 42.72 |
| flatten-sentence | 22.06 | 79.11 | 42.19 | – | 39.82 | 23.46 | 80.93 | 44.78 | – | 41.84 | 23.80 | 78.42 | 46.29 | – | 42.44 |
| raptor | 20.62 | 82.13 | 37.70 | – | 37.55 | 22.42 | 82.21 | 41.45 | – | 39.95 | 24.73 | 83.57 | 44.46 | – | 42.50 |
| *Ours* | | | | | | | | | | | | | | | |
| Bisection | 22.61 | 84.13 | 42.58 | – | 40.98 | 23.54 | 82.53 | 45.87 | – | 42.74 | 24.44 | 80.01 | 47.12 | – | 43.49 |
| DISRetrieval | 22.40 | 79.54 | 44.30 | – | 41.04 | 24.19 | 83.03 | 45.29 | – | 42.58 | 25.20 | 82.51 | 48.51 | – | 44.52 |
| **GPT-4.1-mini + OpenAI Embedding** | | | | | | | | | | | | | | | |
| flatten-chunk | 24.12 | 79.46 | 43.19 | – | 41.13 | 26.90 | 83.09 | 45.45 | – | 43.34 | 27.90 | 80.65 | 48.29 | – | 44.78 |
| flatten-sentence | 25.34 | 79.35 | 46.75 | – | 43.08 | 27.13 | 82.19 | 49.12 | – | 45.28 | 26.94 | 83.30 | 49.60 | – | 45.78 |
| raptor | 23.54 | 78.85 | 43.36 | – | 40.88 | 26.08 | 81.21 | 45.95 | – | 43.26 | 25.94 | 80.59 | 47.40 | – | 43.85 |
| *Ours* | | | | | | | | | | | | | | | |
| Bisection | 24.99 | 81.78 | 46.12 | – | 43.12 | 25.37 | 82.60 | 48.41 | – | 44.54 | 27.81 | 80.48 | 49.97 | – | 45.69 |
| DISRetrieval | 25.04 | 83.57 | 46.68 | – | 43.47 | 26.96 | 82.82 | 49.45 | – | 45.47 | 27.14 | 82.52 | 50.43 | – | 45.97 |
| **Deepseek-v3 + SBERT** | | | | | | | | | | | | | | | |
| flatten-chunk | 17.23 | 81.70 | 35.27 | – | 35.29 | 17.50 | 85.65 | 38.86 | – | 37.99 | 20.11 | 85.71 | 42.48 | – | 40.57 |
| flatten-sentence | 17.74 | 80.26 | 36.99 | – | 36.51 | 18.58 | 81.86 | 41.61 | – | 39.41 | 20.12 | 83.04 | 43.45 | – | 40.82 |
| raptor | 15.75 | 80.27 | 35.03 | – | 34.66 | 17.76 | 82.06 | 38.53 | – | 37.30 | 19.73 | 83.86 | 41.66 | – | 39.77 |
| *Ours* | | | | | | | | | | | | | | | |
| Bisection | 18.05 | 78.76 | 38.83 | – | 37.28 | 18.24 | 85.40 | 41.76 | – | 39.80 | 19.60 | 82.38 | 44.10 | – | 40.96 |
| DISRetrieval | 17.18 | 83.93 | 38.22 | – | 37.44 | 18.10 | 85.40 | 42.92 | – | 40.28 | 19.65 | 85.71 | 44.70 | – | 41.86 |
| **Deepseek-v3 + OpenAI Embedding** | | | | | | | | | | | | | | | |
| flatten-chunk | 19.20 | 82.14 | 39.06 | – | 38.14 | 21.81 | 82.96 | 42.75 | – | 40.78 | 21.77 | 84.62 | 44.90 | – | 42.26 |
| flatten-sentence | 20.45 | 81.70 | 41.02 | – | 39.27 | 22.03 | 82.06 | 44.73 | – | 41.57 | 23.23 | 84.68 | 45.70 | – | 43.04 |
| raptor | 17.47 | 80.89 | 39.55 | – | 37.84 | 20.63 | 82.88 | 43.10 | – | 40.50 | 21.56 | 84.82 | 44.74 | – | 42.17 |
| *Ours* | | | | | | | | | | | | | | | |
| Bisection | 20.62 | 81.86 | 40.73 | – | 39.27 | 22.20 | 84.51 | 44.15 | – | 41.81 | 23.13 | 84.89 | 47.09 | – | 43.57 |
| DISRetrieval | 20.82 | 81.61 | 42.13 | – | 40.01 | 21.95 | 83.93 | 44.73 | – | 42.10 | 23.06 | 84.96 | 45.90 | – | 43.21 |

