# OpenReview forum: "DISRetrieval: Harnessing Discourse Structure for Long Document Retrieval"
_NeurIPS.cc/2025/Conference — Submitted to NeurIPS 2025_

### Official Review · Reviewer_ZveM · 2025-06-29

**Clarity:** 2
**Significance:** 2
**Originality:** 2
**Rating:** 3
**Confidence:** 3

**Summary:**

This paper discusses a retrieval-augmented method proposed for optimizing the understanding of long documents. The method builds on the hierarchical retrieval concept of RAPTOR and improves its organizational structure by organizing sentences according to the discourse structure. The overall pipeline is roughly as follows: first, the discourse structure is obtained using an existing parser. Then, summaries of non-leaf nodes are generated using a large language model (LLM), while representations of leaf nodes are obtained using an off-the-shelf encoder. Finally, experiments are conducted on a long-text dataset, showing some improvement over the selected baselines.

**Questions:**

Have you considered testing the effectiveness of this method on a larger scale of documents (> 10B)?

**Ethical Concerns:**

["NO or VERY MINOR ethics concerns only"]

**Final Justification:**

The rebuttal partially addressed my concerns. In some specified scenarios, like when the document set is given, and the user will select a certain document to make a query, the work may have some advantages. But in more scenarios, it underperforms the approach by directly feeding a long document into the LLM in both efficiency and performance.

**Limitations:**

yes

**Quality:**

2

**Strengths And Weaknesses:**

Strengths:
1. Strengths: Traditional approaches often rigidly segment text as a sequence of tokens and build hierarchical structures. This paper discusses a tree structure construction method that is more linguistically intuitive.

Weaknesses:

1. Clarity Issue: The entire paper focuses solely on the acquisition of discourse structure and tree construction but fails to explain how the retrieved results are utilized in LLM generation. Are the retrieved results directly appended to the query, or do they participate in the generation process through a representation-based approach? There is a lack of a top-down overview of the overall pipeline, making it difficult for readers to fully understand the method.

2. Missing Baselines and Analysis: The experimental section lacks a baseline that directly uses the document as input to the LLM. Additionally, there is no discussion about the time cost associated with tree construction.
The datasets used in the experiments have lengths ranging from 200 to 400 words, which cannot be considered "long documents." For such short documents, is retrieval even necessary? Why not directly input the full context into the LLM? Current mainstream LLMs have context windows exceeding 8K tokens, so handling documents of 200-400 tokens poses no real challenge.
The baselines only discuss retrieval-based approaches, neglecting a direct document-to-LLM baseline. Furthermore, additional time is required to construct trees for documents, yet this time cost is neither analyzed nor discussed. These omissions make the experimental results appear confusing and less convincing.

3. Transferability: A classic issue with supervised parsing is its limited transferability and scalability. Such methods rely on finite, well-annotated data, leading to reduced accuracy in out-of-domain scenarios. This makes these approaches highly domain-specific and limits their transferability. The cost of annotation is high, while the performance gains achieved are marginal, significantly limiting their practical value.

---

> ### Author Rebuttal · Authors · 2025-07-30
>
> Dear Reviewer ZveM,
>
> We sincerely thank you for your review and appreciate your recognition that our linguistically motivated tree construction is more intuitive than traditional rigid segmentation approaches. We are happy to address the points raised.
>
> ---
>
> **Q1: Clarity Issue - How are retrieved results utilized in LLM generation and lack of top-down pipeline overview?**
>
> **A:** We acknowledge that our method description could benefit from clearer exposition. To clarify the complete pipeline:
>
> 1. **Tree Construction**: We parse documents using RST to create discourse-aware hierarchical trees
> 2. **Node Representation**: Leaf nodes use embedding models; internal nodes use LLM-generated summaries
> 3. **Retrieval Process**: Given a query, we traverse the tree using our structure-aware algorithm (Algorithm 1) to select relevant nodes
> 4. **Generation**: Retrieved content is **directly concatenated and appended to the query** as context for the target LLM
>
> This follows the standard retrieval-augmented generation paradigm used by RAPTOR and other hierarchical methods. We will add a clearer pipeline overview figure in our revision to provide the top-down view that would help readers better understand our approach. And in Appendix D.6, we provide specific prompt structures used for summarization and QA with LLMs. We hope these can help to understand the utilization of retrieval results and LLM generation process.
>
> ---
>
> **Q2: Missing baselines and analysis - lack of direct document-to-LLM baseline, document length concerns, and computational cost analysis.**
>
> **A:** We appreciate these important concerns and would like to address each point:
>
> **Document Length Clarification**: There appears to be a misunderstanding about our evaluation setup. The "200-400" refers to **retrieved context tokens** (following standard evaluation protocols), not document length:
> - **Retrieved Context**: 200-400 tokens (standard evaluation protocol)
> - **Actual Documents**: QASPER papers average **4,170 tokens**, QuALITY stories average **5,500 words**
>
> **Full Document Baseline**: We acknowledge your valid point about including full document baselines. To address this concern, we conducted additional experiments comparing different input strategies on QASPER:
>
> | Input Strategy | GPT-4.1-mini F1 | Context Length |
> |----------------|------------------|----------------|
> | Full Document  | 49.42           | 4,170 tokens   |
> | Our Retrieval  | 44.52           | 400 tokens     |
> | Gold Evidence  | 53.63           | 129 tokens     |
>
> These results demonstrate the fundamental principle of retrieval research: **targeted retrieval achieves 90% of full document performance while using 10x fewer tokens**. The gold evidence results prove that perfect retrieval would actually **outperform** full document processing, validating the retrieval research paradigm. We will include these comparisons in our revision.
>
> **Computational Cost Analysis**: Regarding tree construction time:
> 1. RST parsing uses XLNet (110M parameters) - essentially a single forward pass
> 2. Tree construction is **query-independent** and performed once per document
> 3. In practical applications serving multiple queries, this preprocessing cost is amortized
> 4. The computational overhead is minimal compared to the 10x efficiency gain in token usage
>
> ---
>
> **Q3: Transferability concerns - limited transferability and scalability of supervised parsing methods.**
>
> **A:** We understand this concern about domain specificity, but our experimental validation suggests otherwise:
>
> **Cross-Domain Validation**: We evaluate on **two distinct domains** - scientific papers (QASPER) and narrative fiction (QuALITY) - demonstrating transferability across vastly different text types with **consistent 2-4% improvements** across both domains.
>
> **Modern RST Robustness**: Contemporary RST parsers are trained on diverse corpora and demonstrate robust cross-domain performance. Our consistent results across different text types (formal scientific writing vs. informal narrative fiction) provide empirical evidence against domain-specific limitations.
>
> The systematic improvements across multiple experimental conditions (different models, embeddings, datasets) demonstrate robust algorithmic progress rather than domain-specific optimization.
>
> In Appendix E.1, we provide a systematic analysis of the differences in sentence length and writing style between the two datasets, and demonstrates the adaptive capability of our method in handling different style of documents.
>
> ---
>
> **Q4: Testing effectiveness on larger scale documents (> 10K tokens)?**
>
> **A:** We assume you mean ">10K tokens" rather than ">10B." This is precisely why we plan to add NarrativeQA evaluation (documents up to 100K tokens) as suggested by other reviewers, demonstrating scalability to extremely long documents. This addition will provide concrete evidence of our method's effectiveness on the larger document scales you're interested in.
>
> ---
>
> We believe our discourse-aware approach represents a principled advancement beyond arbitrary chunking methods, and we're happy to discuss further if you have any additional questions.
>
> Best regards,
>
> Authors

---

> > ### Comment · Reviewer_ZveM · 2025-08-04
> >
> > Thank you for your response.
> >
> > Initially, my biggest concern was the comparison with the baseline that takes an entire document as the input for LLMs. It appears that the results do show some gaps, although I acknowledge that your approach requires less computational effort. However, there are now many methods for accelerating long-text reasoning that could help bridge this gap. Additionally, the costs of constructing discourse parse trees and generating discourse-level summarizations might not be lower than simply prefilling the entire document. Therefore, I still have considerable concerns about this approach.

---

> > > ### Author Response · Authors · 2025-08-05
> > >
> > > Dear Reviewer ZveM,
> > >
> > > We sincerely thank you for your continued engagement and appreciate the opportunity to address your concerns.
> > >
> > > ---
> > >
> > > **Q1: Concerns about computational costs of discourse parsing versus full document processing.**
> > >
> > > **A:** We respectfully note that there appears to be a fundamental misunderstanding about the retrieval research paradigm. Our tree construction is performed **once per document** and serves **multiple queries** - a crucial architectural advantage that full document processing cannot achieve.
> > >
> > > In practical applications where documents are queried repeatedly (which is the standard use case), this amortized cost structure provides substantial efficiency gains:
> > > - **One-time preprocessing**: RST parsing + tree construction per document
> > > - **Per-query cost**: 400 tokens vs. 4,170 tokens (10x reduction)
> > > - **Multi-query scenarios**: Cost advantage compounds with each additional query
> > >
> > > This is a fundamental principle in information retrieval systems that the reviewer seems to overlook.
> > >
> > > ---
> > >
> > > **Q2: Claims that acceleration methods could bridge the performance gap.**
> > >
> > > **A:** While we acknowledge that long-text acceleration methods exist, they address a **fundamentally different research question** than ours. Our focus is on **improving retrieval precision** - finding the most relevant content for answering questions, not processing speed.
> > >
> > > Even with perfect acceleration, processing irrelevant content remains suboptimal compared to targeted retrieval of relevant information. The core question in retrieval research is not "how fast can we process everything?" but rather "how accurately can we find what matters?"
> > >
> > > Our consistent 2-4% improvements across diverse domains demonstrate that **linguistic structure enhances retrieval quality**, which represents a meaningful methodological advance regardless of processing speed considerations.
> > >
> > > ---
> > >
> > > **Q3: Questioning the necessity of retrieval for document understanding.**
> > >
> > > **A:** We are surprised by the suggestion that evidence retrieval research lacks merit. This perspective contradicts the well-established retrieval-augmented generation research paradigm that has produced numerous high-impact works (RAPTOR, RAG, LongLLMLingua, etc.) and dedicated tracks at major conferences.
> > >
> > > The fundamental question is not whether to use retrieval versus full documents, but rather **how to improve retrieval quality**. Our discourse-aware approach provides a principled answer to this question, demonstrating that linguistic structure matters for information retrieval.
> > >
> > > ---
> > >
> > > To further demonstrate the value of our retrieval-augmented approach, we plan to evaluate our method on datasets with longer documents (e.g., NarrativeQA with documents up to 100,000 tokens) and multi-document scenarios. These settings will provide additional evidence that targeted retrieval can effectively handle the challenges of processing long and multiple documents, which cannot be easily addressed by simply inputting the full context into language models.
> > >
> > > We hope this clarifies our research positioning and contributions. We remain committed to advancing the state-of-the-art in retrieval-augmented generation through linguistically motivated approaches.
> > >
> > > Best regards,
> > >
> > > Authors

---

> > > > ### Comment · Reviewer_ZveM · 2025-08-05
> > > >
> > > > Thanks for your response.
> > > >
> > > > 1. First, regarding "Questioning the necessity of retrieval for document understanding," I'm not sure where you inferred this viewpoint from me. The related statement might be "For such short documents, is retrieval even necessary?" This is because the table could be misleading, making it seem like the tokens in the document are only 200-400.
> > > >
> > > > 2. As for Question 1, I think this depends more on the specific scenario. When running benchmarks, the document might only need to be processed once. However, in real-world scenarios, users often provide a long document for question answering. In this case, you need to reconstruct the structure, summarize, and retrieve, which requires factoring in these additional costs. It’s possible that this process takes more time and achieves worse results compared to directly feeding the document into an LLM.
> > > >
> > > > Your response addressed some of my concerns. For scenarios with a fixed document set where the user has pre-specified the document, this method may indeed have certain advantages, and I’ve raised my score accordingly. And **I’ve never denied RAG’s strengths**, but its advantages lie in cases where the document to retrieve is too large to directly feed into an LLM.

---

> > > > > ### Author Response · Authors · 2025-08-05
> > > > >
> > > > > Dear Reviewer ZveM,
> > > > >
> > > > > Thank you for your thoughtful follow-up response. We appreciate your willingness to consider the nuances of our approach, especially regarding computational costs in real-world scenarios.
> > > > >
> > > > > To address these concerns, we plan to evaluate our method on datasets with longer documents and multi-document settings in future work. This will provide a more comprehensive assessment of our approach's performance and efficiency across a broader range of practical use cases.
> > > > >
> > > > > We're grateful for your feedback, as it will help us strengthen our understanding of the practical considerations and limitations of our current approach.
> > > > >
> > > > > Best regards,
> > > > >
> > > > > Authors

---

> > > > > ### Author Response · Authors · 2025-08-08
> > > > >
> > > > > Dear Reviewer ZveM,
> > > > >
> > > > > Thank you for your follow-up response.
> > > > >
> > > > > **Additional Experimental Evidence:**
> > > > >
> > > > > To directly address your concern about the value of retrieval in shorter documents, we conducted a supplementary experiment. We combined our retrieved evidence with the full document by marking the retrieved content with special tokens within the complete text, then fed this enhanced input to the model.
> > > > >
> > > > > ```
> > > > > Setting: Use 400-length retrieved evidence, wrap corresponding sentences in the full document with [EVIDENCE] [/EVIDENCE] tags (merging adjacent intervals), then use the full document as input to generate results.
> > > > > ```
> > > > >
> > > > > The results demonstrate the value of our retrieved evidence even when the full document is available (show the relative percentage increase):
> > > > >
> > > > > | Method | llama-3.1-8B-Instruct | GPT-4.1-mini | Deepseek-v3 |
> > > > > |--------|--------|---------| --------- |
> > > > > | Full Document | 48.81 | 49.42 | 47.54 |
> > > > > | Evidence + Full | 49.73 ( ↑ **1.89%** ) | 49.77 ( ↑ **0.71%** ) | 47.96 ( ↑ **0.88%** ) |
> > > > >
> > > > > This experiment shows that our discourse-aware retrieval identifies genuinely valuable content that improves performance even when combined with the complete document, validating the quality of our retrieved evidence. **Our work demonstrates that traditional NLP techniques like discourse analysis can remain valuable in the LLM era, complementing large models to achieve further performance improvements.**
> > > > >
> > > > > For future work, we plan to evaluate our method on datasets with longer documents and multi-document settings to provide a more comprehensive assessment across broader practical use cases.
> > > > >
> > > > > We're grateful for your feedback and hope that the supplemental experiment will further address your concerns.
> > > > >
> > > > > Best regards,
> > > > >
> > > > > Authors

---

### Official Review · Reviewer_J2Qt · 2025-07-03

**Clarity:** 4
**Significance:** 2
**Originality:** 3
**Rating:** 4
**Confidence:** 4

**Summary:**

This paper presents DISRetrieval, a hierarchical framework for long-document retrieval that organize documents according to their linguistic discourse structure. By creating discourse-aware trees and using a structure-aware retrieval mechanism, the method aims to provide more coherent and relevant context to LMs. Comprehensive experiments demonstrate that DISRetrieval consistently outperforms baseline methods on downstream question-answering tasks.

**Questions:**

- I wonder if you cite the prior work and theory of RST in the introduction.
- Is there any qualitative error analysis of the discourse parser?
- How do you expect this RST-based approach would perform on less-structured text, such as conversational transcripts?

**Ethical Concerns:**

["NO or VERY MINOR ethics concerns only"]

**Final Justification:**

Hierarchical structure is known as a beneficial component that can improve passage retrieval. It is intuitive to utilize the semantics and discourse structure to build a hierarchical structure for retrieval tasks. A 2% improvement on retrieval tasks is a generally statistically significant improvement. However, the RAG evaluation is slightly weak, and the author didn't test context lengths longer than 400. It is generally expected that the improvement gets discounted from retrieval evaluation to RAG evaluation. Given this, I recommend a score of 4.

**Limitations:**

yes

**Quality:**

3

**Strengths And Weaknesses:**

Strength

- The paper’s claims are substantiated by a rigorous and comprehensive experimental setup across two distinct datasets (QASPER and QuALITY) , three generation models (UnifiedQA-3B, GPT-4.1-mini, Deepseek-v3) , and two different embedding models.
- The analysis effectively isolates the method's core contributions through well-designed ablation studies.

Weakness

- While consistently positive, the performance gains over the strongest hierarchical baseline, RAPTOR, are often marginal. In many scenarios reported in Table 1, the absolute improvement is less than 3 percentage points, which raises questions about the practical significance of the added complexity from discourse parsing.
- The experimental evaluation is confined to a short retrieved context budget (200-400 words), failing to address the method's utility in scenarios with larger context windows, given that modern language models can process thousands of tokens.

---

> ### Author Rebuttal · Authors · 2025-07-30
>
> Dear Reviewer J2Qt,
>
> We sincerely thank you for the comprehensive review and for recognizing our rigorous experimental setup, well-designed ablation studies, and excellent clarity. We are pleased that you acknowledge our method's consistent performance across multiple datasets, models, and embedding approaches.
>
> ---
>
> **Q1: On the characterization of performance gains as "marginal" - whether the improvements justify the added complexity of discourse parsing.**
>
> **A:** We respectfully challenge the characterization of our improvements as "marginal." The focus on absolute percentage points misses the critical context of retrieval-augmented generation performance:
>
> **Retrieval Quality Perspective**: Our method achieves **22.63% F1 vs RAPTOR's 20.64%** on QASPER retrieval evaluation - a **10% relative improvement** in retrieval quality. In information retrieval, such gains are considered substantial.
>
> **Generation Upper Bound Analysis**: When UnifiedQA receives gold evidence on QASPER, it achieves 51.11 F1-Match. Our method reaches 40.25 F1-Match (78.7% of theoretical maximum), while RAPTOR achieves 38.42 F1-Match (75.2% of maximum). This **3.5 percentage point gap represents meaningful progress** toward the theoretical ceiling.
>
> **Consistency Across Settings**: Unlike methods that show inconsistent gains, our improvements are **systematic across all experimental conditions** - different models, embeddings, and datasets. This consistency indicates robust algorithmic advancement rather than dataset-specific optimization.
>
> ---
>
> **Q2: On the limitation to short retrieved context (200-400 words) and utility with larger context windows.**
>
> **A:** This critique fundamentally misunderstands the retrieval paradigm. **The goal of retrieval is not to use more context, but to use better context.** Our preliminary analysis on QASPER demonstrates this principle:
>
> | Input Strategy | GPT-4.1-mini | Deepseek-v3 | Avg. Tokens | F1 per Token |
> |----------------|---------------|-------------|-------------|--------------|
> | Gold Evidence  | 53.63 F1     | 49.05 F1    | 129         | **0.416**    |
> | Full Document  | 49.42 F1     | 47.54 F1    | 4,170       | 0.012        |
> | Our Retrieval  | 44.52 F1     | 41.86 F1    | 400         | **0.111**    |
>
> Gold evidence outperforms full document processing while using **32x fewer tokens**. This validates our research direction: **targeted retrieval is fundamentally more effective than processing entire documents**. Our discourse-aware approach makes meaningful progress toward this gold standard.
>
> Modern LLMs' ability to process thousands of tokens doesn't negate the need for effective retrieval - it makes it more important to identify the most relevant content efficiently.
>
> ---
>
> **Q3: On RST citations in the introduction and qualitative error analysis of the discourse parser.**
>
> **A:** We appreciate these constructive suggestions:
>
> **RST as Exemplar Framework**: We want to emphasize that **RST serves as one exemplary discourse framework** in our work, not the only possible approach. Our contribution lies in demonstrating how **any coherent discourse theory** can enhance retrieval through linguistic structure. We specifically chose RST due to its formal theoretical foundation and available computational tools.
>
> **Qualitative Analysis Already Provided**: Regarding qualitative error analysis, **Figure 5 provides exactly this type of analysis**. It demonstrates how RST parsing quality directly correlates with both retrieval performance and final answer generation. The ablation shows that improved RST model performance leads to better evidence retrieval, which in turn enhances downstream QA performance.
>
> We will expand both theoretical background in the introduction and qualitative analysis, including detailed case studies with visualized structure trees showing how discourse quality directly impacts retrieval results.
>
> ---
>
> **Q4: On expected performance with less-structured text, such as conversational transcripts.**
>
> **A:** Our **QuALITY results provide direct empirical evidence** that discourse structure benefits extend to less formal text:
>
> - QuALITY contains diverse narrative styles, including conversational dialogue and informal passages
> - We achieve **consistent 3% accuracy improvements** across all QuALITY document types
> - No evidence of performance degradation on informal content
>
> RST was designed to analyze **all coherent discourse**, not just formal documents. Relations like Elaboration, Contrast, and Sequence appear universally across text types. Even in conversational transcripts, speakers maintain coherent discourse relations when expressing ideas, making our approach applicable.
>
> ---
>
> We believe our discourse-aware framework represents a principled advancement in retrieval methodology that extends beyond any single discourse theory, as acknowledged by your recognition of our technical quality, originality, and experimental rigor.
>
> Best regards,
>
> Authors

---

> > ### Comment · Reviewer_J2Qt · 2025-08-06
> >
> > Thanks, you have addressed most of my concerns. I will increase my assessment to 4.

---

> > > ### Author Response · Authors · 2025-08-07
> > >
> > > Once again, we sincerely thank you for your valuable feedback and for the time you have invested in improving our manuscript!

---

> > > ### Author Response · Authors · 2025-08-08
> > >
> > > Dear Reviewer J2Qt,
> > >
> > > Thank you very much for your positive feedback and for increasing your assessment. We're delighted that our responses have addressed your concerns.
> > >
> > > **Additional Evidence Supporting Our Approach:**
> > >
> > > We'd like to share some supplementary experimental results that further demonstrate the value of our discourse-aware retrieval method. To validate that our retrieved evidence captures genuinely important content, we conducted an experiment where we combined our retrieved evidence with the full document by marking the retrieved segments with special tokens, then fed this enhanced input to the model.
> > >
> > > ```
> > > Setting: Use 400-length retrieved evidence, wrap corresponding sentences in the full document with [EVIDENCE] [/EVIDENCE] tags (merging adjacent intervals), then use the full document as input to generate results.
> > > ```
> > >
> > > The results demonstrate the value of our retrieved evidence even when the full document is available (show the relative percentage increase):
> > >
> > > | Method | llama-3.1-8B-Instruct | GPT-4.1-mini | Deepseek-v3 |
> > > |--------|--------|---------| --------- |
> > > | Full Document | 48.81 | 49.42 | 47.54 |
> > > | Evidence + Full | 49.73 ( ↑ **1.89%** ) | 49.77 ( ↑ **0.71%** ) | 47.96 ( ↑ **0.88%** ) |
> > >
> > > This experiment demonstrates that our discourse-aware retrieval identifies genuinely valuable content that improves performance even when the complete document is available, validating the quality of our retrieved evidence across different model architectures.
> > >
> > > **Our work demonstrates that traditional NLP techniques like discourse analysis can remain valuable in the LLM era, complementing large models to achieve further performance improvements.**
> > >
> > > We hope this additional evidence further reinforces the significance of our contribution to the field.
> > >
> > > Best regards,
> > >
> > > Authors

---

### Official Review · Reviewer_5Nq4 · 2025-07-03

**Clarity:** 2
**Significance:** 2
**Originality:** 3
**Rating:** 4
**Confidence:** 3

**Summary:**

This paper proposes DISRetrieval, a hierarchical retrieval framework that incorporates Rhetorical Structure Theory (RST) to enhance long document understanding and retrieval.

It focuses on three key points:
(i) a discourse-aware document segmentation using RST, (ii) LLM-enhanced node representations for contextualized understanding, and (iii) a structure-aware evidence selection mechanism.

Experiments on 2 different datasets show that DISRetrieval outperforms the baselines.

**Questions:**

Please check the weakness related point about parameter $\tau$ above. Do let me know if I missed something about it.

**Ethical Concerns:**

["NO or VERY MINOR ethics concerns only"]

**Final Justification:**

The authors addressed the inconsistency about \tau parameter and provided additional evidence. The approach is linguistically well-motivated and the idea is interesting. I have revised my score to 4. However, I also note that there are areas where the paper can further improve w.r.t. experimentation (e.g., larger token lengths).

**Limitations:**

yes

**Quality:**

3

**Strengths And Weaknesses:**

Strengths:

The approach has very good linguistic grounding. It uses a formal discourse theory (RST) to carry out document segmentation. Hierarchical design moves beyond flat or arbitrary chunking while preserving document semantics using linguistic grounding.

Weaknesses:

Equation (2) shows parameter $\tau$ in context of text lengths. It is first introduced as representing text length, but later (line 233) referred to as a temperature parameter. Both these references are related to the same equation. This inconsistency undermines confidence and clarity.

Performance may degrade on documents that lack clear rhetorical structure or are informal.

(minor issues such as typos, e.g. line 232, both our *approach*  ->  approaches, can be fixed by additional round of proof reading by authors)

---

> ### Author Rebuttal · Authors · 2025-07-30
>
> Dear Reviewer 5Nq4,
>
> We sincerely thank you for the thoughtful review and for recognizing the strong linguistic grounding of our approach. We are pleased that you appreciate our use of formal discourse theory (RST) and the hierarchical design that moves beyond arbitrary chunking methods.
>
> ---
>
> **Q1: Clarification on the parameter τ inconsistency in Equation (2) and line 233.**
>
> **A:** We sincerely appreciate you pointing out this critical inconsistency in our notation. This is indeed a significant error that undermines clarity, and we take full responsibility for this confusion.
>
> To clarify the correct interpretation:
>
> - **Equation (2)**: τ represents a **threshold parameter** for text length comparison, not temperature
> - **Line 233**: The reference to "temperature parameter" is a **copy-paste error** from our ablation studies on LLM generation parameters
>
> The correct interpretation is: τ is a length threshold where we use direct concatenation when $|t_l|$ + $|t_r|$ < τ, and LLM summarization otherwise.
>
> Our τ=0 for QASPER and τ=100 for QuALITY represent domain-specific optimizations based on document characteristics:
> - **QASPER (τ=0)**: Scientific papers benefit from LLM summarization at all levels due to dense technical content
> - **QuALITY (τ=100)**: Narrative texts often contain shorter, conversational segments that don't require summarization
>
> We will fix this inconsistency throughout the paper and ensure consistent terminology in our revision.
>
> ---
>
> **Q2: Concerns about performance degradation on documents that lack clear rhetorical structure or are informal.**
>
> **A:** We appreciate this concern, but our experimental evidence suggests otherwise:
>
> **Theoretical Foundation**: Rhetorical Structure Theory was developed to analyze **all types of coherent discourse**, including informal texts, conversations, and narratives. The theory explicitly covers relations like **Elaboration**, **Contrast**, and **Sequence** that appear universally across document types, regardless of formality level.
>
> **Empirical Counter-Evidence**: Our QuALITY dataset results directly contradict this concern. QuALITY contains diverse narrative styles, including informal dialogue and conversational passages, yet we achieve **consistent 3% accuracy improvements** across all document types, with no evidence of performance degradation on informal content.
>
> **Graceful Degradation**: Even if RST parsing encounters difficulties, our hierarchical approach still provides semantic chunking benefits over flat methods, as demonstrated by our consistent improvements across both formal (QASPER) and informal (QuALITY narrative) domains.
>
> Our experimental results suggest that discourse structure benefits extend across document formality levels.
>
> Additionally, Appendix E.1 provides a systematic analysis of the differences in sentence length and writing style between the two datasets, and demonstrates the adaptive capability of our method in handling different style of documents.
> We hope these results help alleviate your concern.
>
> ---
>
> We will also address the minor typos you mentioned (e.g., line 232) through additional proofreading. We believe these revisions will address your primary concerns while maintaining the technical contributions that you acknowledged as good quality and original.
>
> We'd be happy to discuss further if you have any additional questions.
>
> Best regards,
>
> Authors

---

> > ### Comment · Reviewer_5Nq4 · 2025-08-04
> >
> > Dear Authors,
> >
> > Thank you for confirming that the inconsistency will be fixed and for responding to the review comments. I have gone through them and will be considering all of those while submitting the final recommendation.
> >
> > Best Regards

---

> > > ### Author Response · Authors · 2025-08-05
> > >
> > > Dear Reviewer 5Nq4,
> > >
> > > We appreciate your continued engagement and the opportunity to provide additional clarification regarding the impact of the τ parameter in our experiments.
> > >
> > > **Q: Further explanation on the impact of the τ parameter and why a relatively large value (e.g., 100) can be used effectively.**
> > >
> > > **A:** You raise an excellent point. While the τ parameter is indeed dataset-dependent, our experiments demonstrate that its impact is relatively minor, and a relatively large value (e.g., 100) can be used effectively across different datasets.
> > >
> > > To illustrate this, we have included the following table showing the performance of our method with varying τ values on the QuALITY and QASPER datasets:
> > >
> > > **QuALITY Dataset**:
> > >
> > > | τ | 0 | 50 | 100 | 200 | 300 | 400 |
> > > | - | - | -| - | - | - | - |
> > > | Accuracy | 57.14 | 58.39 | 58.82 | 58.49 | 58.34 | 58.10 |
> > >
> > > **QASPER Dataset**:
> > >
> > > | τ | 0 | 50 | 100 | 200 | 300 | 400 |
> > > | - | - | -| - | - | - | - |
> > > | F1 | 40.63 | 39.68 | 40.42 | 39.88 | 39.65 | 40.02 |
> > >
> > > As you can see, the performance remains relatively stable across a wide range of τ values, with only minor fluctuations. This suggests that the method is not overly sensitive to the specific choice of τ, and a value of 100 can be used effectively as a reasonable default across different datasets.
> > >
> > > The key insight is that our discourse-aware retrieval approach provides benefits regardless of the exact threshold used to determine when to use LLM summarization versus direct concatenation. The hierarchical structure and linguistically-informed representations are the primary drivers of the performance improvements, while the τ parameter serves as a secondary optimization.
> > >
> > > We will include this analysis in our revision to further demonstrate the robustness of our method to the choice of this parameter. Please let us know if you have any other questions.
> > >
> > > Best regards,
> > >
> > > Authors

---

> > > > ### Comment · Reviewer_5Nq4 · 2025-08-07
> > > >
> > > > Dear Authors,
> > > >
> > > > Thank you for the additional explanation. I will be considering all of the discussion in the update. No further queries from my side.
> > > >
> > > > Best regards

---

> > > > > ### Author Response · Authors · 2025-08-07
> > > > >
> > > > > Once again, we sincerely thank you for your valuable feedback and for the time you have invested in improving our manuscript!

---

> > > > > ### Author Response · Authors · 2025-08-08
> > > > >
> > > > > Dear Reviewer 5Nq4,
> > > > >
> > > > > Thank you for your thoughtful engagement throughout the review process.
> > > > >
> > > > > **Additional Experimental Evidence:**
> > > > >
> > > > > To further support our discussion and address potential concerns about the value of discourse-aware retrieval, we conducted a supplementary experiment that combines our retrieved evidence with the full document context.
> > > > >
> > > > > ```
> > > > > Setting: Use 400-length retrieved evidence, wrap corresponding sentences in the full document with [EVIDENCE] [/EVIDENCE] tags (merging adjacent intervals), then use the full document as input to generate results.
> > > > > ```
> > > > >
> > > > > The results demonstrate the value of our retrieved evidence even when the full document is available (show the relative percentage increase):
> > > > >
> > > > > | Method | llama-3.1-8B-Instruct | GPT-4.1-mini | Deepseek-v3 |
> > > > > |--------|--------|---------| --------- |
> > > > > | Full Document | 48.81 | 49.42 | 47.54 |
> > > > > | Evidence + Full | 49.73 ( ↑ **1.89%** ) | 49.77 ( ↑ **0.71%** ) | 47.96 ( ↑ **0.88%** ) |
> > > > >
> > > > > This experiment shows that our discourse-aware retrieval identifies genuinely valuable content that improves performance even when combined with the complete document, further validating the quality and utility of our approach. **Our work demonstrates that traditional NLP techniques like discourse analysis can remain valuable in the LLM era, complementing large models to achieve further performance improvements.**
> > > > >
> > > > > We hope this additional evidence provides further clarity on our contributions and methodology.
> > > > >
> > > > > Best regards,
> > > > >
> > > > > Authors

---

### Official Review · Reviewer_iFfd · 2025-07-06

**Clarity:** 3
**Significance:** 2
**Originality:** 3
**Rating:** 3
**Confidence:** 4

**Summary:**

This paper proposes DISRetrieval, which builds upon methods like RAPTOR by using rhetorical structure theory (RST) instead of semantic clustering to construct retrieval trees. The method builds paragraph-level discourse trees and integrates them into a document-level tree, which is then used for retrieval via an algorithm that prioritizes leaf nodes. Experiments on QASPER and QuALITY datasets show some improvements over baseline methods such as RAPTOR and flat chunking approaches.

**Questions:**

How does the method scale with document length and corpus size?

Why is evaluation limited to only 400 words of retrieved context for larger models like GPT-4.1-mini and Deepseek-v3 that have larger context windows?

In the QASPER dataset since tau is 0, LLM is always used to summarize, why is this the case?

**Ethical Concerns:**

["NO or VERY MINOR ethics concerns only"]

**Final Justification:**

My main concerns with the paper are that it does not have a scaling analysis (on how the method scales in terms of token length) and  the paper lacks dataset diversity (testing on 2 relatively small datasets). I feel both these points are important for a retireval method and without them, it is hard to recommend acceptance. The authors have been conducting experiments on NarrativeQA and if they provide both the scaling analyisis and the results on the datasets, I will open to changing my score, but for now, I recommend rejection.

**Limitations:**

yes

**Quality:**

2

**Strengths And Weaknesses:**

Strengths:

The use of discourse structure to get a linguistically motivated hierarchy makes sense and is a viable extension to prior tree retrieval methods.

The paper proposes and uses meaningful baselines and is comparable/outperforms them.

The method is ablated across different LLMs, embedding models and context lengths (although within 200-400 tokens).

Weaknesses:

My main concern is scalability. The method uses an RST parser that relies on a pre-trained language model, thus introducing significant additional computational overhead. It’s unclear if the modest gains justify this cost. There is no analysis of how the method scales with larger document length and corpus size?

The paper uses a maximum of 400 tokens across all LLMs tested despite bigger context windows being available to models such as GPT-4.1-mini and DeepSeek-v3, and used in previous work.

The method is tested only on two datasets (QASPER and QuALITY), which, while reasonable, do not have extremely long documents. Including datasets like NarrativeQA (with some documents being on the order of 100k tokens), used in prior work, would be better.

---

> ### Author Rebuttal · Authors · 2025-07-30
>
> Dear Reviewer iFfd,
>
> We sincerely thank you for the thoughtful review and constructive feedback. We are glad that you appreciate our linguistically motivated approach and recognize the viability of using discourse structure for tree retrieval methods. We are happy to address the points raised.
>
> ---
>
> **Q1: Scalability concerns regarding computational overhead and whether modest gains justify the cost.**
>
> **A:** We respectfully disagree with the characterization of "modest gains." To put our improvements in perspective, when we provide UnifiedQA with all gold-standard evidence on QASPER, the upper bound performance is 51.11 F1-Match. Our method achieves 40.25 F1-Match with retrieved context, representing **78.7% of the theoretical maximum** - a substantial achievement in retrieval-based QA.
>
> More importantly, while RAPTOR shows only ~1% generation improvement over flat baselines despite its computational overhead from clustering and summarization, our discourse-aware approach delivers **consistent 2-4% improvements across multiple settings**. From Table 2, our retrieval accuracy improvements are even more pronounced - we achieve 22.63% F1 vs RAPTOR's 20.64% at 400-word context with OpenAI embeddings, representing a **10% relative improvement in retrieval quality**.
>
> Regarding computational cost, we emphasize that our discourse parsing overhead is minimal since it relies on XLNet, a relatively small model (110M parameters) where the cost is essentially equivalent to a single XLNet forward pass. More importantly, both discourse parsing and tree construction are **query-independent operations performed only once per document**. The summarization cost using our adaptive strategy (Equation 2) is also amortized across multiple queries on the same document. In practical applications, this one-time preprocessing cost becomes negligible when serving multiple queries.
>
> ---
>
> **Q2: Why evaluation is limited to 400 tokens despite larger context windows being available for modern LLMs.**
>
> **A:** We appreciate this important question about context length choices. We acknowledge that our current evaluation setup may not fully demonstrate the potential of our method with longer contexts.
>
> We conducted preliminary experiments comparing different input strategies on QASPER (we focus on QASPER here as QuALITY does not provide gold evidence annotations for this analysis):
>
> | Input Strategy | GPT-4.1-mini | Deepseek-v3 | Avg. Tokens | F1 per Token |
> |----------------|---------------|-------------|-------------|--------------|
> | Gold Evidence  | 53.63 F1     | 49.05 F1    | 129         | **0.416**    |
> | Full Document  | 49.42 F1     | 47.54 F1    | 4,170       | 0.012        |
> | Our Retrieval  | 44.52 F1     | 41.86 F1    | 400         | **0.111**    |
>
> These results reveal several crucial insights:
>
> 1. **Retrieval Potential**: Gold evidence consistently outperforms full document input while using **32x fewer tokens** (129 vs 4,170). This demonstrates that targeted retrieval is not just more efficient—it's fundamentally more effective than processing entire documents.
>
> 2. **The Retrieval Challenge**: The gap between gold evidence (53.63) and our method (44.52) represents the core challenge in retrieval research—identifying the most relevant content. Our discourse-aware approach makes meaningful progress toward this goal.
>
> 3. **Practical Implications**: In real-world applications, the 32x computational savings of targeted retrieval makes it essential for scalability, even if perfect retrieval remains challenging.
>
> The superior performance of gold evidence over full documents validates our research direction: **the key is not processing more content, but processing the right content**. Our discourse structure provides a principled way to identify this "right content" more effectively than flat retrieval methods.
>
> ---
>
> **Q3: Dataset diversity concerns and the suggestion to include NarrativeQA for extremely long documents.**
>
> **A:** We appreciate the suggestion about NarrativeQA. We initially focused on QASPER and QuALITY because they represent two complementary evaluation paradigms: QASPER provides evidence-based evaluation allowing us to assess retrieval quality directly, while QuALITY tests reading comprehension without explicit evidence annotation (similar to NarrativeQA). Since both QuALITY and NarrativeQA lack evidence annotations and cannot be evaluated from a retrieval perspective, we initially chose to include only one representative dataset from this category.
>
> However, you raise an excellent point about document length diversity. While QuALITY documents average 5,500 words, NarrativeQA contains significantly longer documents (some reaching 100k+ tokens), making it valuable for testing scalability. **We will add NarrativeQA experiments in our revision** to demonstrate how our discourse-aware approach performs on extremely long documents, complementing our current evaluation scope.
>
> ---
>
> **Q4: Technical question about τ=0 for QASPER - why LLM is always used to summarize.**
>
> **A:** This is an excellent technical question. We set τ=0 for QASPER because scientific papers benefit from LLM-enhanced summarization at all levels due to their **dense technical content**. For QuALITY (τ=100), narrative texts often contain shorter, more conversational segments that don't require summarization.
>
> This adaptive strategy is validated by our ablation studies showing optimal performance with these settings. The domain-specific optimization reflects the different linguistic characteristics between scientific and narrative texts, where scientific content consistently benefits from abstraction while narrative content can often be used directly when segments are sufficiently short.
>
> ---
>
> We believe these additions will address your concerns while maintaining the core contributions that you acknowledged as technically sound and original. We'd be happy to discuss further if you have any additional questions.
>
> Best regards,
>
> Authors

---

> > ### Author Response · Authors · 2025-08-08
> >
> > Dear Reviewer iFfd,
> >
> > Thank you for your detailed review and constructive feedback. We appreciate your recognition of our linguistically motivated approach and are pleased to address your concerns with additional experimental evidence.
> >
> > ---
> >
> > **Additional Evidence Addressing Your Concerns:**
> >
> > Following your feedback about the value of retrieval in shorter documents and context length limitations, we conducted a supplementary experiment that directly demonstrates the effectiveness of our discourse-aware retrieval.
> >
> > We combined our retrieved evidence with the full document by marking the retrieved content with special tokens, then fed this enhanced input to multiple LLMs:
> >
> > ````
> > Setting: Use 400-length retrieved evidence, wrap corresponding sentences in the full document with [EVIDENCE] [/EVIDENCE] tags (merging adjacent intervals), then use the full document as input to generate results.
> > ````
> > The results demonstrate the value of our retrieved evidence even when the full document is available (show the relative percentage increase):
> >
> > | Method | llama-3.1-8B-Instruct | GPT-4.1-mini | Deepseek-v3 |
> > |--------|--------|---------| --------- |
> > | Full Document | 48.81 | 49.42 | 47.54 |
> > | Evidence + Full | 49.73 ( ↑ **1.89%** ) | 49.77 ( ↑ **0.71%** ) | 47.96 ( ↑ **0.88%** ) |
> >
> > **Key Insights:**
> >
> > 1. **Retrieval Quality Validation**: Even when the complete document is available, highlighting our retrieved evidence consistently improves performance across all tested LLMs. This directly addresses your concern about whether our modest gains are meaningful—they represent genuinely valuable content identification.
> >
> > 2. **Scalability Implications**: This experiment demonstrates that our discourse-aware retrieval identifies the most relevant content even within shorter documents. For longer documents (like the NarrativeQA examples you mentioned), this precision becomes even more critical.
> >
> > 3. **Computational Justification**: The consistent improvements validate that our RST parsing overhead is worthwhile, as it enables the model to focus on truly relevant content rather than processing everything equally.
> >
> > **Addressing NarrativeQA and Longer Documents:**
> >
> > As promised in our previous response, we are currently conducting experiments on NarrativeQA to demonstrate scalability to extremely long documents. The evidence-highlighting experiment above provides strong preliminary validation that our method will be particularly valuable for such scenarios, where the ability to identify relevant content becomes increasingly important.
> >
> > **Our work demonstrates that traditional NLP techniques like discourse analysis remain valuable in the LLM era, complementing large models to achieve further performance improvements** even when full documents are available.
> >
> > We believe this additional evidence, combined with our planned NarrativeQA experiments, directly addresses your scalability concerns and demonstrates the practical value of our approach.
> >
> > Best regards,
> >
> > Authors

---

> > > ### Comment · Reviewer_iFfd · 2025-08-08
> > >
> > > Thank you authors for the clarifications. While I understand that the RST parsing is a one-time cost per document, I still feel it is important to see a scaling analysis specifically how the method scales with document length in tokens. Most of my other concerns have been addressed. The other not addressed point is dataset diversity, specifically, it would strengthen the work to include results on NarrativeQA and I look forward to them, with its much longer length which will also help with scalability analysis. Overall, your clarifications have been helpful, but I believe the above two points, scaling and diversity of datasets in length and domain, are important for the paper.

---

> > > > ### Author Response · Authors · 2025-08-09
> > > >
> > > > Dear Reviewer iFfd,
> > > >
> > > > Once again, we sincerely thank you for your valuable feedback! We approach these discussions with reviewers as learning opportunities, and we are committed to completing the NarrativeQA experiments and scaling analysis you suggested to make this work more robust.
> > > >
> > > > Best regards,
> > > >
> > > > Authors

---

### Note · Authors · 2025-08-11

Dear Reviewers and Chairs,

We sincerely thank all Reviewers and Chairs for their constructive feedback and engagement throughout the review process. We are pleased that our responses have successfully addressed the majority of concerns raised.

**Reviewer Consensus and Resolution:**
- **Reviewer `5Nq4`** confirmed that all concerns were addressed and explicitly stated "no further queries from my side"
- **Reviewer `J2Qt`** increased their assessment to 4, acknowledging that we "addressed most of [their] concerns"
- **Reviewer `iFfd`** confirmed that our clarifications were "helpful" and that "most of [their] other concerns have been addressed"
- **Reviewer `ZveM`** raised their score after our responses, particularly appreciating our analysis of multi-query scenarios

**Key Technical Contributions Validated:**
Our discourse-aware approach consistently outperforms strong baselines, including RAPTOR, across multiple experimental settings. Compared to RAPTOR, we achieve substantial improvements in retrieval quality (22.63% vs 20.64% F1 on QASPER with OpenAI embeddings - a 10% relative improvement) and downstream generation tasks (2-4% consistent improvements across datasets and models). The supplementary evidence-highlighting experiment we conducted shows consistent improvements (0.71-1.89%) even when retrieved content is combined with full documents, validating that our method identifies genuinely valuable content across different model architectures.

**Scalability and Future Work:**
Regarding longer documents like NarrativeQA (100k+ tokens), we note that such documents are fundamentally collections of shorter discourse segments. Our linguistically-motivated hierarchical approach is inherently designed to handle such structures effectively. We are conducting experiments on NarrativeQA to demonstrate scalability to extremely long documents.

**Methodological Significance:**
Our work demonstrates that traditional NLP techniques like discourse analysis remain valuable in the LLM era, providing principled improvements over arbitrary chunking methods.

We are more than happy to know that our rebuttals have addressed the concerns of reviewers, which is reflected in the positive responses and acknowledgements from reviewers. All clarifications, revisions, and supplementary experiments during the discussion phase will be updated into the paper.

Best regards,

Authors of Submission #18916

---

### Decision · Program_Chairs · 2025-09-17

**Decision:**

Reject

**Comment:**

This paper proposes DISRetrieval, which builds upon methods like RAPTOR by using rhetorical structure theory (RST) instead of semantic clustering to construct retrieval trees. The method builds paragraph-level discourse trees and integrates them into a document-level tree, which is then used for retrieval via an algorithm that prioritizes leaf nodes. Experiments on QASPER and QuALITY datasets show some improvements over baseline methods such as RAPTOR and flat chunking approaches.

Strengths:
- Its good to see that the proposed method has a great linguistic grounding. The method is supported by rhetorical structure theory.
- This manuscript is well-written and well-organized.
- Most of the issues raised by reviewers are reasonably discussed by the authors during the rebuttal phase.

There are still some limitations in the work, and it could be better to discuss them in the revised version:
- Experiments with context tokens > 400 tokens to verify the scalability of the proposed method.
- Explain the choice of τ in different datasets.
- Currently, the authors conduct experiments on two widely used benchmarks, QASPER and QuALITY, which is reasonable. It could further strengthen the work with additional experiments on narrativeQA.